# AI-guided few-shot inverse design of HDP-mimicking polymers against drug-resistant bacteria

Tianyu Wu[1,7], Min Zhou [2,7], Jingcheng Zou [3], Qi Chen[3], Feng Qian[1], Jürgen Kurths [4,5,6], Runhui Liu [2,3] ✉ & Yang Tang [1] ✉

Host defense peptide (HDP)-mimicking polymers are promising therapeutic alternatives to antibiotics and have large-scale untapped potential. Artificial intelligence (AI) exhibits promising performance on large-scale chemical-content design, however, existing AI methods face difficulties on scarcity data in each family of HDP-mimicking polymers ($<10^2$), much smaller than public polymer datasets ($>10^5$), and multi-constraints on properties and structures when exploring high-dimensional polymer space. Herein, we develop a universal AI-guided few-shot inverse design framework by designing multi-modal representations to enrich polymer information for predictions and creating a graph grammar distillation for chemical space restriction to improve the efficiency of multi-constrained polymer generation with reinforcement learning. Exampled with HDP-mimicking $\beta$-amino acid polymers, we successfully simulate predictions of over $10^5$ polymers and identify 83 optimal polymers. Furthermore, we synthesize an optimal polymer $DM_{0.8}iPen_{0.2}$ and find that this polymer exhibits broad-spectrum and potent antibacterial activity against multiple clinically isolated antibiotic-resistant pathogens, validating the effectiveness of AI-guided design strategy.

As the global risk of antimicrobial resistance continues to escalate, it is urgent to develop alternative strategies to combat antibiotic-resistant bacteria[1–4]. One of the pressing clinical needs is the discovery of promising broad-spectrum antibacterial agents against both Gram-positive and Gram-negative bacteria, especially against antibiotic-resistant pathogens[5,6]. Host defense peptides (HDPs) have garnered considerable attention owing to the advantages of broad-spectrum antibacterial property and low susceptibility to antimicrobial resistance[7,8]. However, the application of HDPs is hindered by their easy enzymatic degradation and expensiveness[9,10]. HDP-mimicking polymers have been designed to address the shortcomings of natural HDPs and have emerged as promising antimicrobial alternatives[11–15]. Furthermore, the discovery of HDP-mimicking antibacterial polymers is limited to conventional designing and optimization strategy, which is semiempirical and inefficient. Artificial intelligence (AI) enables rapid design and optimization of various chemical-contents[16–21], and it is expected to substantially accelerate the discovery of promising HDP-mimicking polymers[22–25].

Nevertheless, two orthogonal challenges inhibit the practical usage of AI for polymers design, specifically in polymer prediction and

[1]Key Laboratory of Smart Manufacturing in Energy Chemical Process, East China University of Science and Technology, Shanghai 200237, China. [2]State Key Laboratory of Bioreactor Engineering, East China University of Science and Technology, Shanghai 200237, China. [3]Shanghai Frontiers Science Center of Optogenetic Techniques for Cell Metabolism, Frontiers Science Center for Materiobiology and Dynamic Chemistry, Key Laboratory for Ultrafine Materials of Ministry of Education, Research Center for Biomedical Materials of Ministry of Education, School of Materials Science and Engineering, East China University of Science and Technology, Shanghai 200237, China. [4]Potsdam Institute for Climate Impact Research (PIK), Potsdam 14473, Germany. [5]Institut für Physik, Humboldt-Universität zu Berlin, Berlin 10115, Germany. [6]The Research Institute of Intelligent Complex Systems, Fudan University, Shanghai 200433, China. [7]These authors contributed equally: Tianyu Wu, Min Zhou. ✉e-mail: rliu@ecust.edu.cn; yangtang@ecust.edu.cn

polymer generation. For polymer prediction, the available few-shot data of peptide-mimicking polymers ($10^2$ or even fewer) in each family is much smaller than data of polymers from public datasets ($10^5$–$10^6$ or even more)[26–28]. This scarcity of data leads to a serious issue for causing overfitting of the predictive model when method transfer, resulting in a decline in performance of the predictive models. For polymer generation, polymer space is constructed by numerous variables of polymers in structures, composition, chain length etc., presenting challenges for AI to efficiently and accurately explore for reasonable polymers with multiple desirable property constraints, of which some may even be inversely related, in the vast high-dimensional polymer space[29]. This challenge implies that the existing AI methods focus more on optimizing polymers with tailored sequence or composition, since these polymers can be coarse-grained simulated or enumerated[30], than exploring for novel chemical structures for subunits[31–33]. Consequently, there is an urgent need to develop an efficient AI method which is capable of predicting and generating novel polymer structures with multi-constraints using few-shot polymer data.

To address those two challenges above, herein we develop an end-to-end AI-guided few-shot inverse design framework to realize an effective exploration of novel polymers under the condition of few-shot data and multiple constraints in high-dimensional polymer space. To enhance the performance of predictive model for polymers, we construct multi-modal polymer representations to enrich the multi-scale polymeric information for few-shot polymer data. This increases the alignment between predictive models and actual polymer systems compared to one single representation[34,35]. To accurately explore for novel polymer structures within desired properties, we develop a grammar knowledge distillation, which distills a graph grammar fragment set according to the existed few-shot polymer data and recombines these grammar as distilled molecules set to restrict the high-dimensional polymer space. These process contributes to improve the efficiency of AI exploration under multi-constraints and ensure the chemical rationality and availability of polymer structures. HDP-mimicking $\beta$-amino acid polymers have attracted significant attention and demonstrated enormous potential for various applications due to striking structural similarity to natural peptides, superior biocompatibility and high resistance to protease hydrolysis[13,36–38]. By implementing our AI design framework, using only 86 HDP-mimicking $\beta$-amino acid polymers as a model[39–43], we successfully simulate predictions of over $10^5$ polymers and indeed identify 83 candidates exhibiting broad-spectrum activity against antibiotic-resistant bacteria. In addition, we synthesize an optimal polymer $DM_{0.8}iPen_{0.2}$ and find that this polymer demonstrates broad-spectrum and potent antibacterial activity against drug-resistant clinically isolated pathogens, which validates the effectiveness and reliability of our AI design method. Furthermore, our framework is a completely data-driven method and it can be universally transferred to various few-shot polymer design tasks. With constructing proper predictive model and generative model, the usage can be further expanded. In one word, AI-guided polymer design accelerates the discovery of potent antimicrobial agents against antibiotic-resistant bacteria and offers a promising strategy to combat antibiotic resistance.

## Results
### Framework overview
The main procedure of our polymer inverse design framework was illustrated in Fig. 1. First, we collected a set of existing data comprising chemical structures and their bioactivity activity of HDP-mimicking $\beta$-amino acid polymers. The chemical structure of the totally 86 polymers was composed of a positively charged subunit (dimethyl (DM), monomethyl (MM)) and a hydrophobic subunit (cyclopentyl (CP), cyclohexyl (CH), etc.) in different proportions with the total chain length of 20 (Fig. 1a and Supplementary Fig. 1). Previous studies indicated that the biological activity of $\beta$-amino acid polymers was mainly

influenced by varying the side chain hydrophobicity (side chain carbon atom number and its atomic spatial arrangement) and the ratio of hydrophobic component/positively charged component. The antibacterial activity data including the minimum inhibitory concentration (MIC) values of polymers against Gram-positive bacterial *Staphylococcus aureus* (*S. aureus*) and Gram-positive bacterial *Escherichia coli* (*E. coli*), as well as hemolytic toxicity data was collected with the minimum concentration to cause 10% hemolysis ($HC_{10}$) values (Supplementary Data 1). Due to the characteristics of abundant structures of $\beta$-amino acid polymers, we conducted a refined classification according to the different position of side chain substituents and cyclic/non-cyclic substitution pattern and defined 11 scaffolds to accurately characterize the polymer structure one-on-one (Fig. 1a, Supplementary Fig. 2, Supplementary Table 1 and Supplementary Data 2). Secondly, we transformed the polymer structure into multi-modal polymer representations to capture comprehensive multi-scale polymer information for training the predictive model so as to enhance the model performance (Fig. 1b). Then, we developed a graph grammar distillation method to pre-train the generative model for generating $\beta$-amino acid polymers structures which tend to rationality and availability based on the chemical principle (Fig. 1c). Specifically, the chemical structure of collected $\beta$-amino acids polymer aforementioned and homologous natural $\alpha$-amino acids were resolved into a variety of molecular graph grammar fragments, which were subsequently recombined to form new molecules (Supplementary Data 3). The process of resolution-recombination was iterated and these recombined mass molecules were used for pre-training the generative model (Supplementary Data 4 and Supplementary Data 5), allowing for the generation of a more focused polymer chemical space. Our graph grammar distillation method could not only contribute to restrict the vast and high-dimensional chemical space of polymers but also generate more reasonable novel polymer structures for practical usage. Finally, we combined these two pre-trained models in reinforcement learning (RL) to form a polymer inverse design framework. The predictive and the generative model were respectively regarded as the environment and the agent to construct a RL pattern. The generative model generated a set of novel polymers and the predictive models provided the corresponding rewards after the evaluation of their bioactivity and structures. Through these rewards, the parameters of the generative model were updated to search for new polymer structures in next RL episode. With such iteration of generative and predictive model, a set of candidate polymers were finally discovered according to the predefined bioactivity values.

### Construction and evaluation of the multi-modal polymer representations
To overcome the limitation associated with the limited information available from few-shot polymers, we employed the multi-modal polymer representations to extract comprehensive multi-scale polymer information. First, we constructed a text-sequence polymer representation using BigSMILES[44] and we further introduced a definition to incorporate information about the proportion of cationic and hydrophobic subunits in the polymer chain, so as to extract the global-level polymer information.

Secondly, we constructed a polymer graph representation by concreting the linking rules of bonding descriptors in BigSMILES syntax, which demonstrated the connection relationships between the subunits in polymer, as new nodes and edges in polymer graph to extract local-level polymer information (see Methods). Finally, we utilized the descriptors from the Mordred calculator[45] to describe the characteristics and properties of cationic and hydrophobic subunits. To ensure as much information as possible are embedded in descriptors, we employed a machine learning-based descriptor downselection process[46], filtering out 40 optimal descriptors with strong correlation to target activity from the original 3654 descriptors (see Methods). In

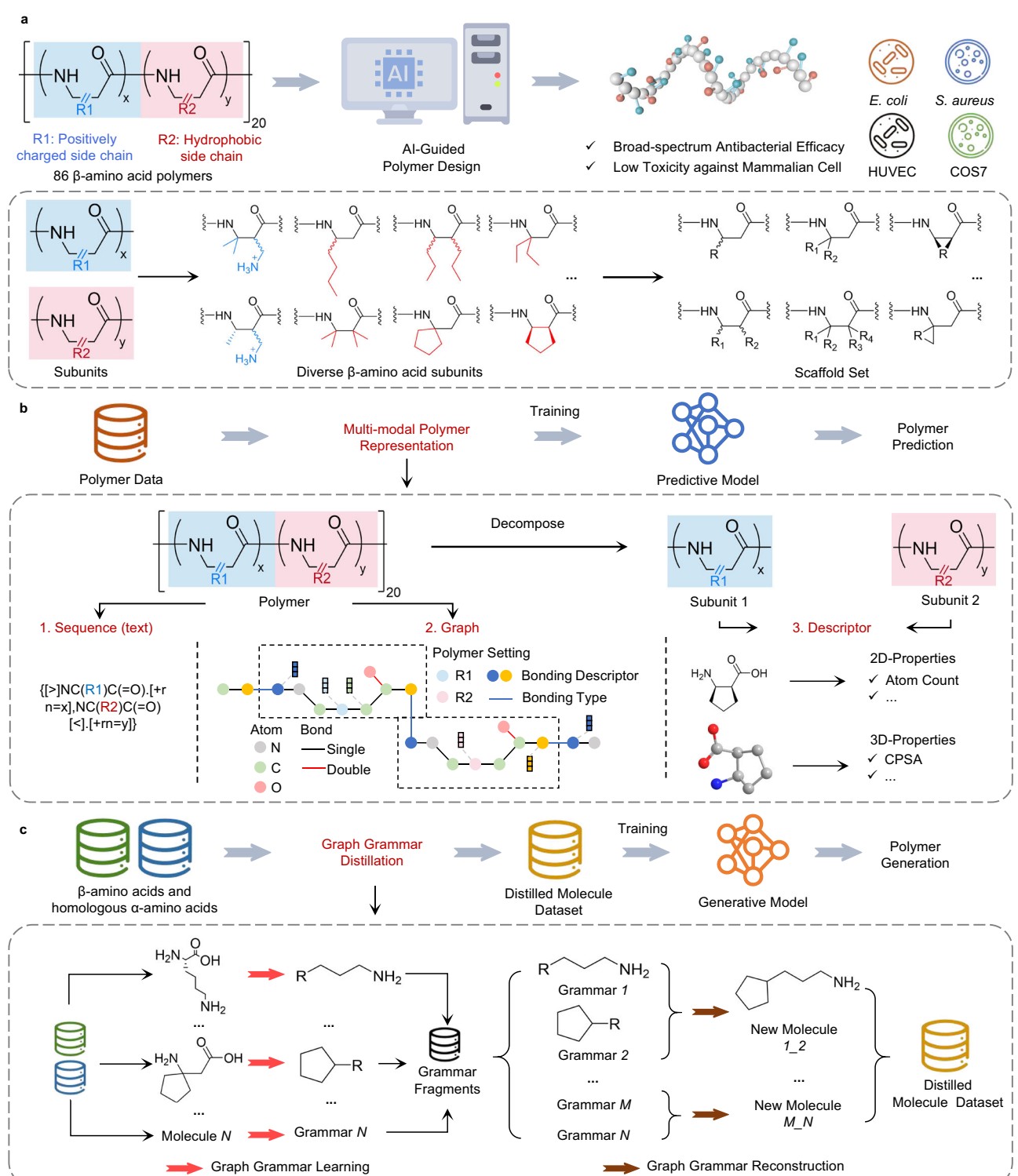

**Fig. 1 | Framework overview. a** By collecting 86 data comprising chemical structures and their bioactivity of *β*-amino acid polymers, we develop an AI-guided few-shot inverse design framework to find promising polymers with broad-spectrum antibacterial efficacy and low cytotoxicity. In addition, we conduct a refined classification according to the different position of side chain substituents and cyclic or non-cyclic substitution pattern, which defines a scaffold set for the following polymer generation. x and y are defined as the percentages of a positively charged subunit and a hydrophobic subunit in *β*-amino acid polymers, respectively. "R₁, R₂, R₃, R₄" means that more than one substitution point should be decorated. Note that all subunits are achiral. **b** We conduct a multi-modal polymer representation method, including text sequence, graph with additional polymer settings and descriptors embedded with 2D- and 3D-properties of subunits to expand for multi-scale polymer information to realize few-shot polymer prediction. **c** We develop a graph grammar distillation method in which we utilize *β*-amino acids and natural *α*-amino acids to learn the split graph grammar fragments. These fragments are reconstructed as distilled molecule dataset to pre-train the generative model to restrict the huge chemical space for exploration.

addition, we conducted the data augmentation on three multi-modal polymer representations by introducing permutation invariance[47], which allowed for significant reductions in prediction errors for multi-component systems (see Methods).

In this manuscript, we randomly selected 80% of collected 86 polymers as the training set $D_{train\_ori}$ and the rest of 20% of data were set as the unseen testing set $D_{test}$. Thus, all models were trained and evaluated in same data situation. We first evaluated the performance of applying descriptor downselection and data augmentation that were two important operations of influencing the input representations. We defined an augmented training data $D_{train\_aug}$, which contained original training data $D_{train\_ori}$ along with additional data by tuning all possible polymer sequences of cationic and hydrophobic subunits in all representations. In this stage, we constructed 4 classic machine learning based regression models, including Gradient Boosting Decision Tree (GBDT)[48], Random Forest (RF)[46], Extreme Gradient Boosting (XGB)[49] and Adaptive Boosting (Adaboost)[50] for bioactivity prediction. The model performance was characterized by calculating the mean R-squared coefficient (R2). We applied a 15-fold cross validation on $D_{train\_ori}$ and $D_{train\_aug}$ to evaluate the performance of different models with fixed descriptors (Fig. 2a–l and Supplementary Fig. 9).

Generally speaking, GBDT models performed best than other methods on each task (Fig. 2a–c for GBDT, Fig. 2d–f for RF, Fig. 2g–i for XGB, Fig. 2j–l for Adaboost). The results showed that the mean R2 values of GBDT for $D_{train\_ori}$ increased gradually to the 0.626, 0.640 and 0.795 on predicting the values of MIC$_{S.aureus}$, MIC$_{E.coli}$ and HC$_{10}$ of polymers with applying descriptors downselection, showing that more related information was selected step by step (Fig. 2a–c, blue boxes). After applying data augmentation, the mean R2 values showed a more obvious increase to 0.739, 0.681 and 0.831 for $D_{train\_aug}$ compared to using $D_{train\_ori}$, indicating the increased prediction accuracy (Fig. 2a–2c, red boxes). Via a final evaluation with GBDT on $D_{test}$, the mean R2 values reached 0.672, 0.537 and 0.834 for MIC$_{S.aureus}$, MIC$_{E.coli}$ and HC$_{10}$, regarding as a machine learning baseline in this manuscript. Results for all machine learning models on $D_{test}$ were demonstrated in Fig. 2m–o.

Moreover, we further studied the performance of all the predictive network by combing three modals of text sequence of polymer, polymer graph and descriptors with applying descriptor downselection and data augmentation discussed before. In addition, we added GBDT, RF, XGB and Adaboost as basic benchmark models and we also introduced the most commonly used polymer representation of Morgan fingerprints[51,52] for comparison. All models were trained on $D_{train\_aug}$ and evaluated on $D_{test}$ for performance comparison, and R2 was again used as the metric. We designed different deep neural network structures for each single representation and an integrated framework for multi-modal representations (see Methods). With final evaluation on $D_{test}$, it was obviously found that GBDT again demonstrated the best in all machine learning based models with mean R2 values of 0.672, 0.537 and 0.834 for MIC$_{S.aureus}$, MIC$_{E.coli}$ and HC$_{10}$ (Fig. 2m–o). The "Descriptor_Opt" demonstrated the best in all single representation with mean R2 values of 0.606, 0.415 and 0.852, whereas the mean R2 values of Morgan was 0.606, 0.415 and 0.852. In addition, the combination of three modals "Seq+Graph+Descriptor_Opt" showed the highest mean R2 values at 0.697, 0.556 and 0.900, indicating that our constructed multi-modal polymer representations obviously improved the accuracy and stability of the predictive model for few-shot polymers (More results concluded in Supplementary Fig. 13 and Supplementary Tables 3, 4).

We further compared in detail about the bioactivity of all polymers between the predictive values and the real measured ones (Fig. 3a). We divided the data according to the positively charged subunit and hydrophobic subunit so as to more rigorously embody the differences of the compositions (Supplementary Fig. 1). Note that log transformation was performed to all the estimated results. The final R2 scores of our model reached 0.91, 0.88 and 0.91 on MIC$_{S.aureus}$, MIC$_{E.coli}$

and HC$_{10}$ for DM series polymers, and 0.92, 0.84 and 0.96 on MM series polymers. It was obviously found from the radar plot that the predicted values highly fit real measured values, indicating that our predictive model was capable of making credible predictions of the bioactivity of β-amino acid polymers.

Moreover, considering the variegation of antibacterial polymers and the rarity of partial types of polymers, we evaluated the transferability of our proposed method in order to broaden its applicability. We collected additional data on α-amino acid polymers[53], polymethacrylates[54–57], polymethacrylamides[58] and other categories[59–61] to evaluate the transferability of our model (Supplementary Data 6). Note that we use the metric of mean absolute error (MAE) to show direct difference of the transferability performance of our model in different categories of antibacterial polymers. According to the evaluated results, for α-amino acid polymers, the MAE was only 0.51 and 0.79 for MIC$_{S.aureus}$ and MIC$_{E.coli}$, which was close to the MAE of β-amino acid polymers (0.17 and 0.40 for MIC$_{S.aureus}$ and MIC$_{E.coli}$, Fig. 3b–e). This fact suggested promising prospects for transferring our method to other categories of antibacterial polymers that possess similar structural characteristics to β-amino acid polymers. For polymethacrylates, the MAE reached 1.24 and 1.95 (nearly six times than β-amino acid polymers) for MIC$_{S.aureus}$ and MIC$_{E.coli}$, respectively (Fig. 3f–i). For polymethacrylamides, the MAE reached 2.33 and 3.75 (nearly ten times than β-amino acid polymers) for MIC$_{S.aureus}$ and MIC$_{E.coli}$, respectively (Fig. 3j–m). These results showed that our model encountered challenges when predicting the properties of other polymers for example polymethacrylates and polymethacrylamides due to substantial dissimilarities with β-amino acid polymers. In summary, our model demonstrated promising transferability to α-amino acid polymers which had highly similarity with our trained data of β-amino acid polymers, while our model were not suggested to be directly transferred to other categories before we further improved the performance of the model (All results shown in Supplementary Figs. 14–22).

## Performance evaluation of graph grammar distillation

We evaluated the performance of the generative model produced from the pre-training of graph grammar distillation using ChEMBL[62] as a control, which was a commonly used dataset to pre-train a generative model and included abundant and diverse chemical structures. We conducted a fine-tuning process with reinforcement learning (RL) for 450 iterations on these two pre-trained generative models to generate polymer subunits with desired chemical structures in multiple given constraints of carbon atoms number and elemental composition in the side chain structure (Task 1 in Method). The generated subunits were scored as reward feedback (details of the rewards see Methods). The subunit that met the given constraints would get a positive reward, and the subunit that did not met the constraints would get a negative reward.

The results showed that in the last several iterations in RL training, the average total rewards of polymers on the values of MIC$_{S.aureus}$ and carbon atom constraints for graph grammar distillation pre-trained generative model got a positive value, indicating that the generated subunits met the design requirements (Fig. 4a). In contrast, the corresponding average total rewards in the ChEMBL pre-trained generative model obtained the negative values (Fig. 4a), indicating that many generated subunits were hard to meet the design requirements, especially for carbon atom constraint (Fig. 4b, c). More comparative results between the model pre-trained by ChEMBL and graph grammar distillation were shown in Supplementary Figs. 25–27. We further evaluated the performance of the graph grammar distillation pre-trained generative model in multi constraints of all three bioactivities, polymer carbon atom number and carbon ring number (Task 2 in Methods, Supplementary Figs. 28, 29). These results exhibited that graph grammar distillation successfully restricted the high-dimensional chemical space and the generative model pre-trained by

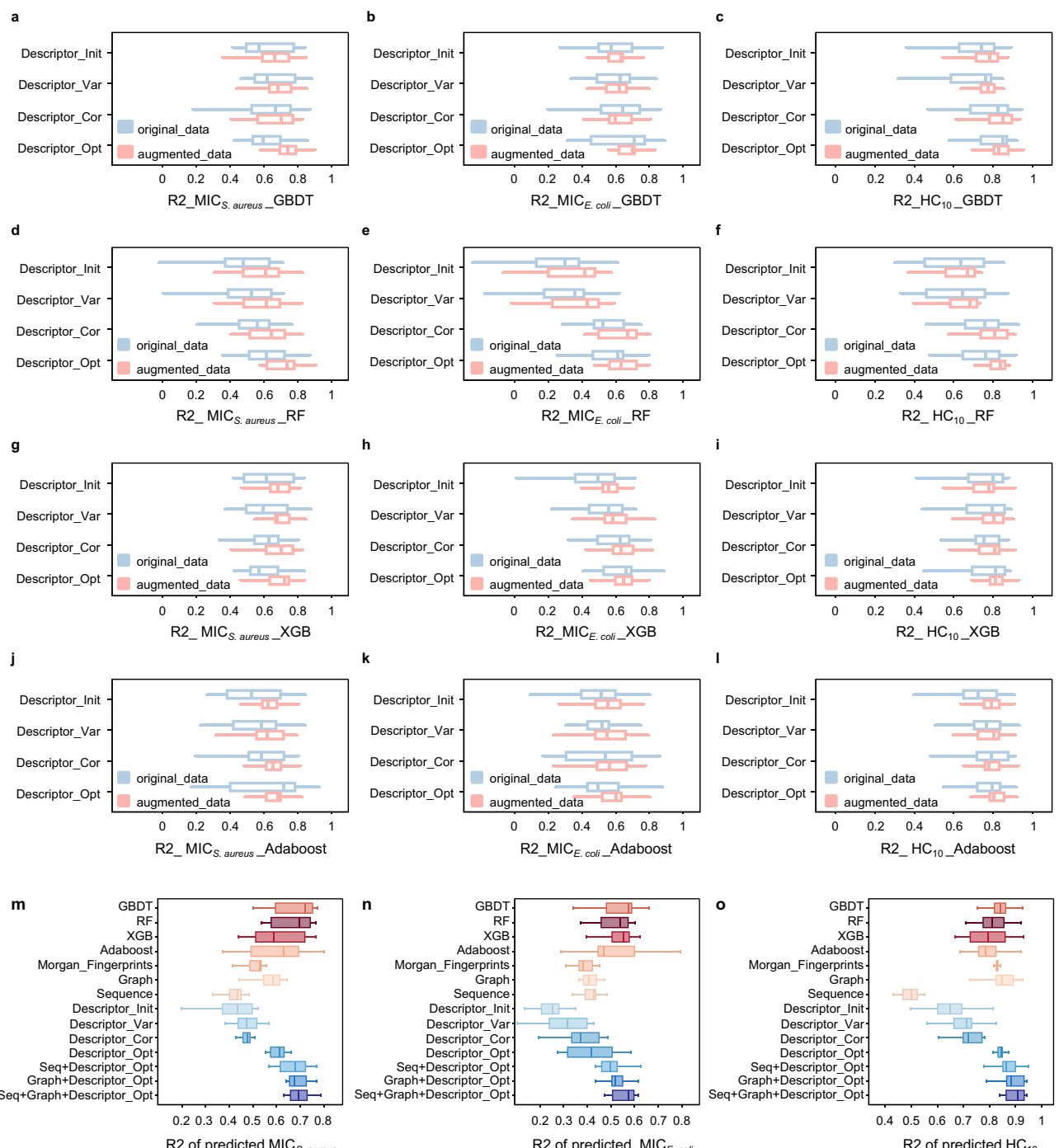

**Fig. 2 | Results of the predictive model.** Cross validation results ($n = 15$ fold) with Gradient Boosting Decision Tree (GBDT, **a**–**c**), Random Forest (RF, **d**–**f**), Extreme Gradient Boosting (XGB, **g**–**i**), Adaptive Boosting (Adaboost, **j**–**l**) for applying descriptors downselection and data augmentation on predicting the values of the minimum inhibitory concentration (MIC) for *S. aureus* ($MIC_{S.aureus}$) and *E. coli* ($MIC_{E.coli}$) and the value of the minimum concentration to cause 10% hemolysis ($HC_{10}$) with the metric of R-squared coefficient ($R^2$). Descriptor_Init to Descriptor_Opt are different sets of descriptors (from the initial set to the optimized set) when downselection. Red boxes are results for augmented data and the bules for original data. The borders of the boxes indicate the first quartile (left) and the third quartile (right) of the results. The line in the box indicates the median. The whiskers refer to the most extreme, nonoutlier data points, with minima on the left and maxima on the right. **m**–**o** Property prediction results of unseen test set $D_{test}$ with deep neural network on $MIC_{S.aureus}$, $MIC_{E.coli}$ and $HC_{10}$ with different polymer representation combination ($n = 10$). The borders of the boxes indicate the first quartile (left) and the third quartile (right) of the results. The line in the box indicates the median. The whiskers refer to the most extreme, nonoutlier data points, with minima on the left and maxima on the right. "Seq" is the abbreviation of "Sequence" (Source data are provided as a Source Data file).

it possessed the strong capabilities for an efficient customized generation of polymer subunits.

To verify the structural diversity of generated polymers by our generative model under multiple constraint conditions, we generated the β-amino acid polymer library consisting of 2114 types of hydrophobic subunits for every cationic subunit and visualized all hydrophobic subunits with Topological Data Analysis Mapper (TMAP)[63]. These hydrophobic subunits covered the possible side chain

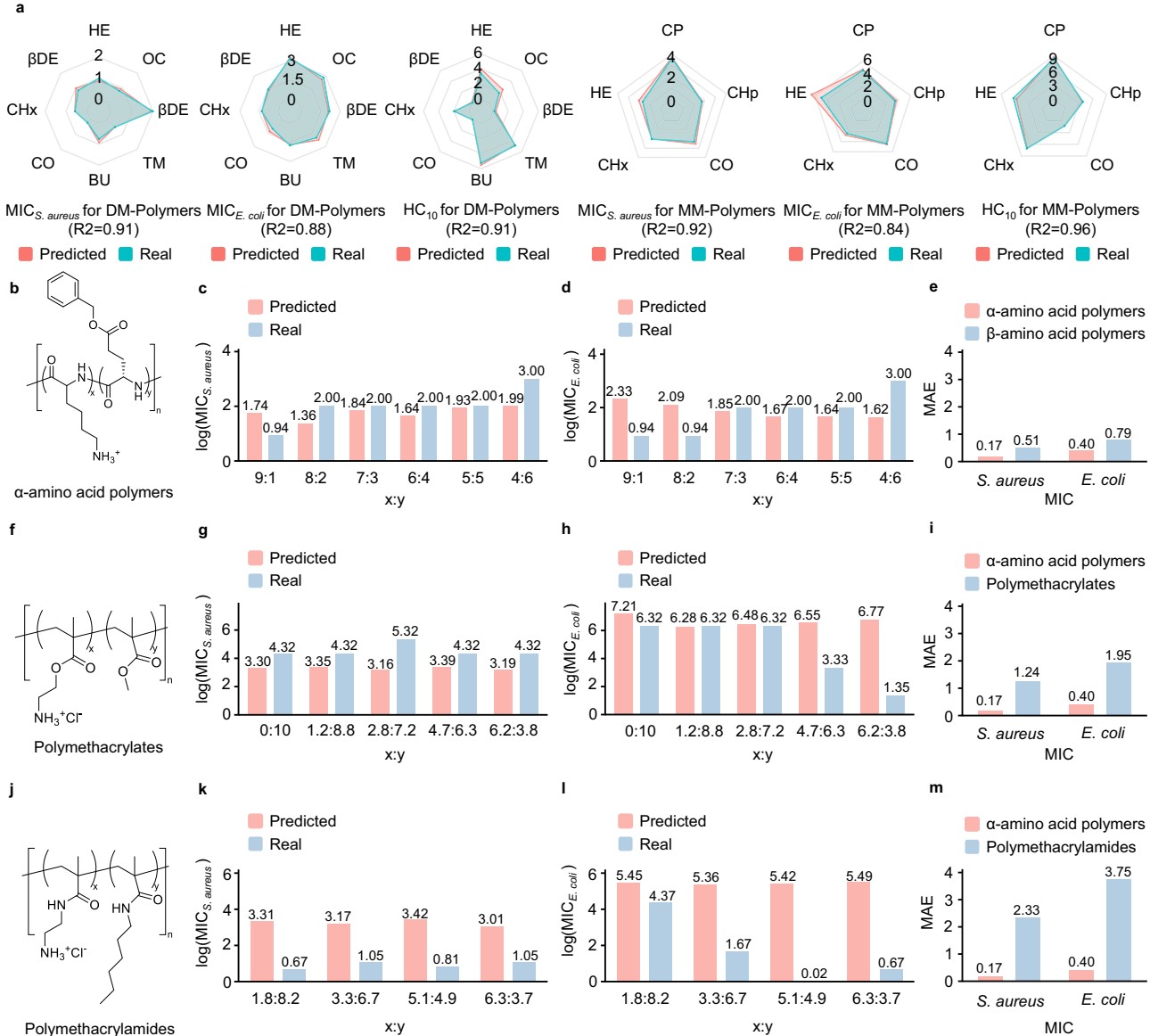

**Fig. 3 | Results of the predictive model. a** Comparison between predicted values and real measured values for β-amino acid polymers. Results show a desirable accuracy, with the metric of R-squared coefficient (R2) reaching 0.91, 0.88 and 0.91 on the values of the minimum inhibitory concentration for *S. aureus* (MIC*S.aureus*), E. coli (MIC*E.coli*) and the value of the minimum concentration to cause 10% hemolysis (HC10) on polymers with dimethyl (DM) subunit, and 0.92, 0.84 and 0.96 on polymers with monomethyl (MM) subunit. Text abbreviations (HE, OC, etc.) mean different hydrophobic subunits. All values are transformed values by natural logarithm. **b–m** Comparison between predicted values and real measured values for α-amino acid polymers (**b**), polymethacrylates (**f**) and polymethacrylamides (**j**). Predicted values on MIC*S.aureus* (**c, g, k**) and MIC*E.coli* (**d, h, l**) in various proportion are recorded with natural logarithmic transformation (log). We also visualize mean absolute error (MAE) of the predictions for each polymer (**e, i, m**) to show the difference when model transferring (Source data are provided as a Source Data file).

structures, encompassing various substitution forms, equally distributed as the defined scaffolds, including the representative six styles of β-amino acid polymers (Fig. 4d). This indicated that our graph grammar distillation based generative model was able to generate various of β-amino acid polymers with abundant cationic and hydrophobic subunits for the discovery of novel antibacterial candidates.

## Visualized analysis of AI-predicted structure and activity of β-amino acid polymers

We made overall predictions on three bioactivities of the generated cationic-hydrophobic β-amino acid polymers with the aforementioned 2114 types of hydrophobic subunits. The ratio of cationic to hydrophobic subunit was limited to 0.1 to 0.9 (9 samples). Thus, the bioactivity data of 19,026 polymers could be generated for each cationic

subunit. Taking the DM/MM as a representative cationic subunit, we visualized three predicted distributions of the bioactivities of MIC*S.aureus*, MIC*E.coli* and HC10 and we further categorized the polymers according to the different ranges of carbon numbers in hydrophobic subunits (Fig. 4e–j). According to the prediction, for polymers with DM subunit, concretely 85.0%, 92.2% and 92.8% of polymers in each range (5-6, 7-8 and 10-11) reached the MIC values < 25 μg mL⁻¹ against *S. aureus* (Fig. 4e) and 44.1%, 36.5% and 28.6% against *E. coli* (Fig. 4f). Whereas, for polymers with MM subunit, less polymers possessed high activity against *S. aureus* and *E. coli* with MIC value < 25 μg mL⁻¹, and 7.2%, 29.7% and 21.7% polymers in each range reached the value against *S. aureus* (Fig. 4h) and 7.5%, 2.1% and 0.0% against *E. coli* (Fig. 4i). These results indicated that the polymers with DM subunit showed greater opportunities to explore the promising broad-spectrum antibacterial polymers. Moreover, for the given threshold of HC10 value > 50 μg mL⁻¹,

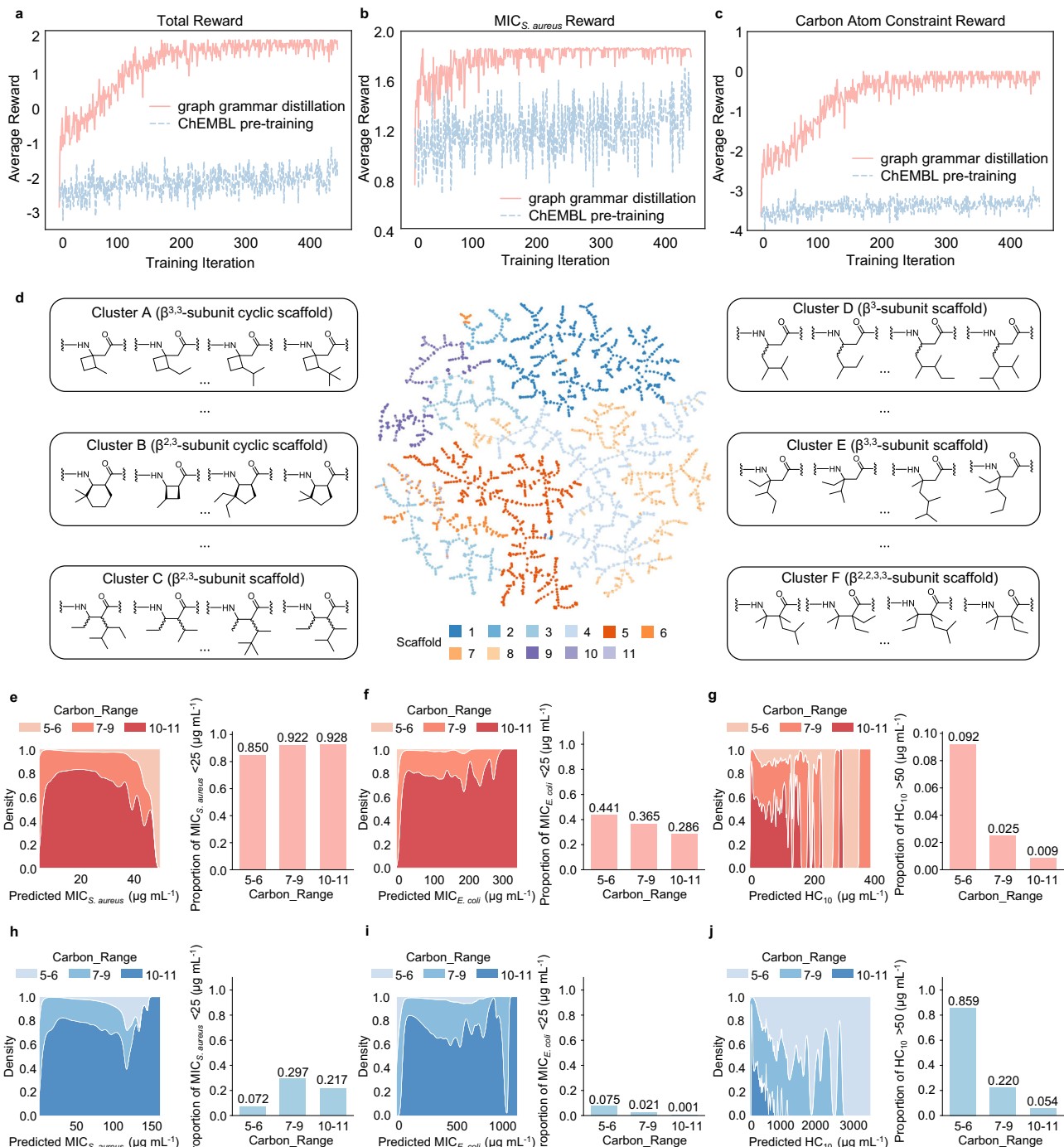

**Fig. 4 | Results of the generative model and visualized analysis. a–c** Average reward curves show opposite model performance when fine-tuning model pre-trained by graph grammar distillation (red) and ChEMBL dataset (blue) with reinforcement learning. Total reward consists of the constraints about the values of the minimum inhibitory concentration for *S. aureus* (MIC$_{S.aureus}$) and number of carbon numbers (less than 11). A higher reward means that the model generates more desired structures as expected. **d** Overview of the Topological Data Analysis Mapper (TMAP) for 2114 generated hydrophobic subunits colored by the corresponding scaffolds. Subunits with the same scaffolds are generally clustered together. Note that all subunits are achiral. Cluster A–E include the representative six styles of β-amino acid polymers which are mostly appeared in our data. **e–j** Property prediction distributions on three bioactivities, including the values of the minimum inhibitory concentration for *S. aureus* (MIC$_{S.aureus}$), *E. coli* (MIC$_{E.coli}$) and the value of the minimum concentration to cause 10% hemolysis (HC$_{10}$), for the generated 19,026 β-amino acid polymers with fixed dimethyl (DM, **e–g**) or monomethyl (MM, **h–j**) positively charged subunits. Ratios of polymers in different carbon range which reach the threshold of specific bioactivity are calculated (Source data are provided as a Source Data file).

no matter what the cationic subunit was DM or MM, the ratios of the generated polymers in 19,026 samples gradually decreased with an increasing carbon number (Fig. 4g, j). The aforementioned findings guided us to select an appropriate range of carbon numbers (<11) for better polymer activity in the following design.

## Visualized analysis of AI-predicted antibacterial selection index (SI) of β-amino acid polymers

We further made overall predictions on antibacterial SI of the generated β-amino acid polymers as one of the important parameters to evaluate selectivity and safety of the antibacterial

agents[64]. Herein, we focused on exploring the optimal antibacterial β-amino acid polymer with a high SI value by finding the suitable hydrophobic subunit using DM as cationic subunit. We used a uniform manifold approximation and projection (UMAP)[65] to project all β-amino acid polymers with DM subunit onto a 2D embedding chemical space (Fig. 5). We collected the SI values of all generated 19,026 polymers by calculating $HC_{10}$/MIC against *S. aureus* and *E. coli*, respectively, and we conducted the classification and visualized analysis on these data according to the range of SI. Finally, we filtered out 9 broad-spectrum antibacterial candidates with both high activities and SI values including 4 different structures of hydrophobic subunits. It was worth noting that most suboptimal polymers (greed, gray and yellow points) were clustered near the candidates (red star), which inspired us to make a detailed exploration for the hidden candidates in the near space of the optimal polymer points.

## Discovery of broad-spectrum antibacterial candidate polymers

We run our framework to discover broad-spectrum antibacterial polymers with desirable bioactivities ($MIC_{S.aureus} < 25 \mu g \ mL^{-1}$, $MIC_{E.coli} < 25 \mu g \ mL^{-1}$ and $HC_{10} > 100 \mu g \ mL^{-1}$). We conducted a systematical exploration for candidate polymers by using different β-amino acid polymer scaffolds (Supplementary Figs. 30–50), and finally found 83 novel broad-spectrum antibacterial candidates by limiting the carbon number of hydrophobic subunit to less than 11 (Supplementary Tables 6–20). Displaying the scaffold of $β^3$-amino acid as a hydrophobic subunit example model (Task 3 in Methods), we collected 640 β-amino acid polymers using DM as the cationic subunit and various substituted $β^3$-amino acids in the RL fine-tuning process (Fig. 6a). We made a prediction on the values of $MIC_{S.aureus}$, $MIC_{E.coli}$ and $HC_{10}$ with the predictive model, and projected all values in a 3D-space with the three properties as coordinates (Fig. 6b). From these results, we finally filtered out 5 candidate polymers in this polymer

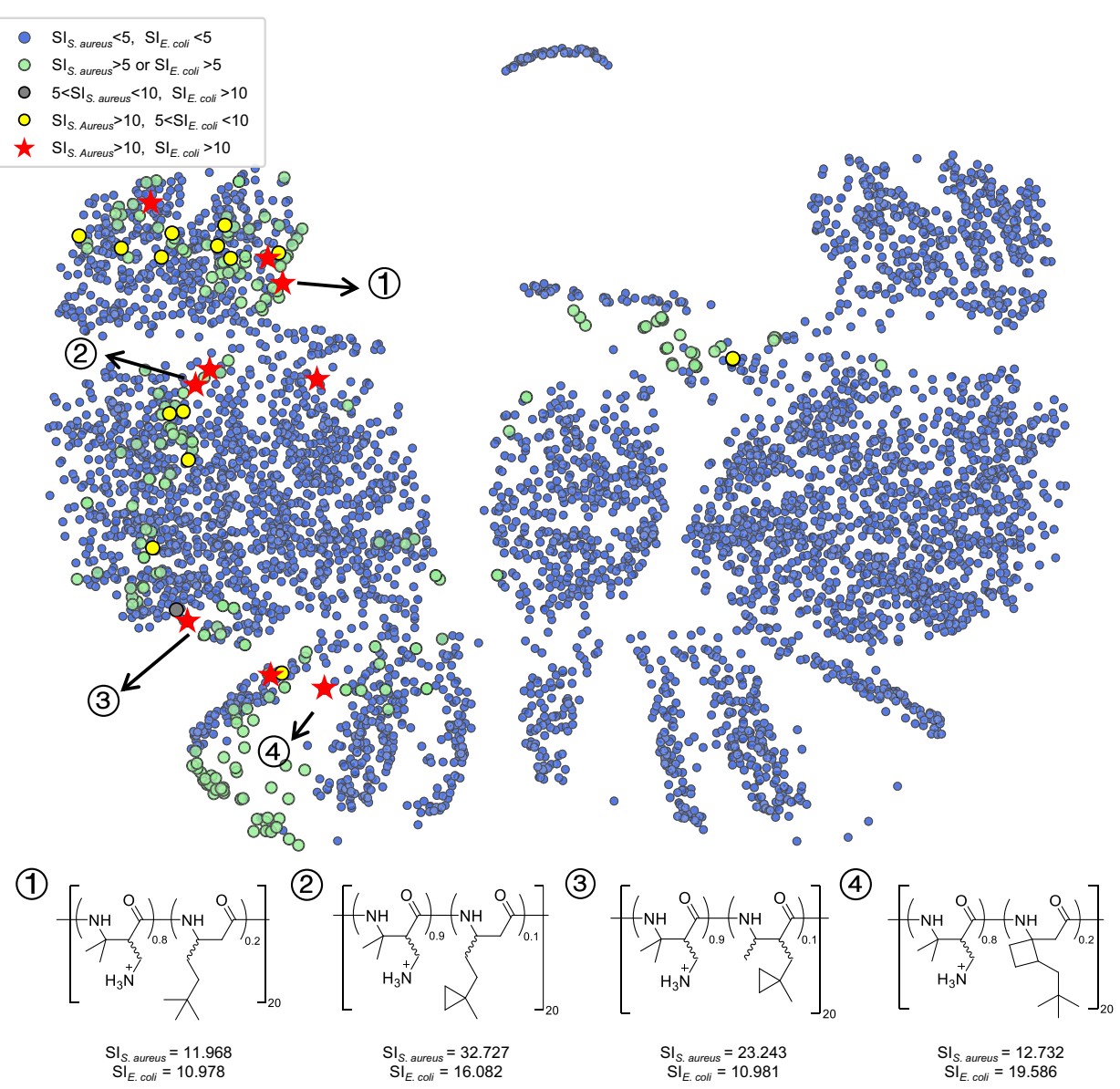

**Fig. 5 | Chemical space visualization with Uniform manifold approximation and projection (UMAP) colored by the selected index (SI) values of generated polymers.** We construct a chemical space with the generated polymers bearing dimethyl (DM) as positively charged subunit. Each polymer is colored according to the values of the SI for *S. aureus* ($SI_{S.aureus}$) and *E. coli* ($SI_{E.coli}$) by the predicted values of the minimum inhibitory concentration for *S. aureus* ($MIC_{S.aureus}$), *E. coli*

($MIC_{E.coli}$) and the value of the minimum concentration to cause 10% hemolysis ($HC_{10}$). Polymers with desirable SI values ($SI_{S.aureus} > 10$ and $SI_{E.coli} > 10$) are displayed with red stars. Moreover, it can be clearly found that most suboptimal polymers with $SI_{S.aureus} > 5$ or $SI_{E.coli} > 5$ (green, yellow and gray points) are clustered together nearby the red star points, meaning that more potential structures exist around them (Source data are provided as a Source Data file).

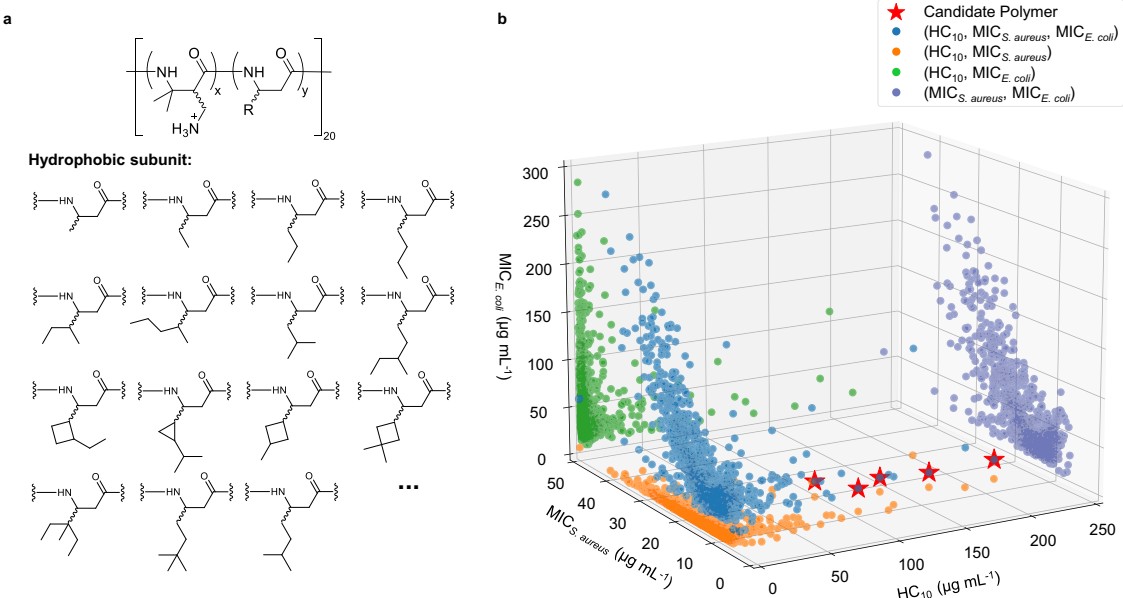

**Fig. 6 | Discovery of broad-spectrum antibacterial polymers bearing $\beta^3$-amino acid. a** Various $\beta^3$-amino acid generated in the discovery process with fixed dimethyl (DM) subunit. Note that all subunits are achiral. x and y are defined as the percentages of a positively charged subunit and a hydrophobic subunit in $\beta$-amino acid polymers, respectively. **b** 3D-projection of the bioactivities on the values of the minimum inhibitory concentration for *S. aureus* (MIC$_{S.aureus}$), *E. coli* (MIC$_{E.coli}$) and the value of minimum concentration to cause 10% hemolysis (HC$_{10}$) for the generated $\beta^3$-amino acid polymers with fixed DM subunit. Blue plots describe the distribution of each generated polymer coordinated with (HC$_{10}$, MIC$_{S.aureus}$ and MIC$_{E.coli}$) according to the predicted value. Orange plots are the projection on (HC$_{10}$, MIC$_{S.aureus}$) space, green plots are the projection on (HC$_{10}$, MIC$_{E.coli}$) space, while purple plots are the projection on (MIC$_{S.aureus}$, MIC$_{E.coli}$) space. Red stars are polymers reaching three desired properties of MIC$_{S.aureus}$ < 25, MIC$_{E.coli}$ < 25 and HC$_{10}$ > 100, simultaneously.

**Table 1 | Display of the final filtered out candidate polymers with predicted properties**

| Structure | Polymer | x:y | HC$_{10}$(µg mL$^{-1}$) | MIC$_{S.aureus}$ (µg mL$^{-1}$) | MIC$_{E.coli}$(µg mL$^{-1}$) |
|---|---|---|---|---|---|
| | Candidate 1 | 9:1 | 119.6 | 8.31 | 24.9 |
| | Candidate 2 | 7:3 | 134.6 | 15.5 | 14.6 |
| | Candidate 3 | 8:2 | 137.7 | 13.9 | 15.2 |
| | Candidate 4 | 9:1 | 214.8 | 12.7 | 20.0 |
| | Candidate 5 | 8:2 | 156.9 | 10.2 | 24.1 |

HC$_{10}$ means the value of the minimum concentration to cause 10% hemolysis, while MIC$_{S.aureus}$ and MIC$_{E.coli}$ mean the values of the the minimum inhibitory concentration for *S. aureus* and *E. coli*.

scaffolds which meet ideal properties (MIC$_{S.aureus}$ < 25, MIC$_{E.coli}$ < 25 and HC$_{10}$ > 100) (Table 1). In addition, we expanded our model on poly($\alpha$-amino acid) and polypeptoid scaffolds to explore further potential application of our method (Supplementary Figs. 51–55). All experimental settings were same, and we also screened out several broad-spectrum antibacterial candidate polymers.

### Synthesis and broad-spectrum antibacterial validation of AI-predicted $\beta$-amino acid polymers
In order to verify the accuracy and reliability of the AI system for predicting antibacterial activity and hemolytic toxicity of HDP-

mimicking $\beta$-amino acid polymers, we selected the $\beta$-amino acid polymers DM$_x$iPen$_y$ from the numerous candidate polymers. Firstly, the DM monomer and *i*Pen monomer were copolymerized and subsequently deprotected to obtain the $\beta$-amino acid polymers with different ratios of positive charge and hydrophobicity (Fig. 7a)[66]. Gel permeation chromatography (GPC) characterization of the *N*-Boc-protected polymers showed a narrow distribution of molecular weight ($D$ = 1.09–1.15) and controllable molecular weight as well as chain length (DP = 20–23) (Fig. 7b and Table 2). Proton nuclear magnetic resonance (¹H NMR) of *N*-Boc-deprotected polymers implied a continuous increase in the proportion of hydrophobic subunit (Fig. 7c).

Then, we tested the hemolytic toxicity against human red blood cells (hRBCs) and cytotoxicity of this polymer libraries using Human umbilical vein endothelial cell line (HUVEC) and the African green monkey kidney fibroblasts (COS7) cells as representative mammalian cells, and found that the hemolytic and cytotoxic activities increased (values decreased) along with the increasing ratio of the $i$Pen component, with the minimum concentration to cause 50% hemolysis ($HC_{50}$) values dropping from 200 $\mu$g mL$^{-1}$ to 12.5 $\mu$g mL$^{-1}$ and the minimum concentration to cause 50% inhibition ($IC_{50}$) values dropping from 200 $\mu$g mL$^{-1}$ to 75 $\mu$g mL$^{-1}$. When the hydrophobicity ratio reached 30%, the hemolysis of the polymers was significant (Fig. 7d–f). In addition, we also tested the antibacterial activity of these polymers against multiple drug-resistant Gram positive and Gram negative bacteria including three strains of methicillin-resistant *S. aureus* (MRSA), clinically isolated multidrug-resistant strains *S. aureus* R03

and two strains of vancomycin resistant enterococcus (VRE), and two strains of multidrug-resistant *Escherichia coli*. All these polymers displayed strong and broad-spectrum antibacterial activities with MIC in the range of 6.25–50 $\mu$g mL$^{-1}$. When the hydrophobicity ratio was in the range of 20–40%, the polymers showed potent activities against all bacterial strains with MIC in the range of 6.25–12.5 $\mu$g mL$^{-1}$ (Table 3). Combining the experimental data of hemolysis, cytotoxicity and antibacterial activities, DM$_{0.8}$$i$Pen$_{0.2}$ as the optimal antibacterial candidate exhibited broad-spectrum and potent antibacterial activity, which was consistent to our results predicted by AI system. Our AI system made accuracy predictions on antibacterial activity, and also found out the cationic/hydrophobic subunit ratio with low toxicity. Furthermore, DM$_{0.8}$$i$Pen$_{0.2}$ showed desirable antibacterial selectivity with SI values against mammalian cells of hRBC, HUVEC and COS7 at 12–32 (Fig. 7g–i), and antibacterial SI values bigger than 10 indicated

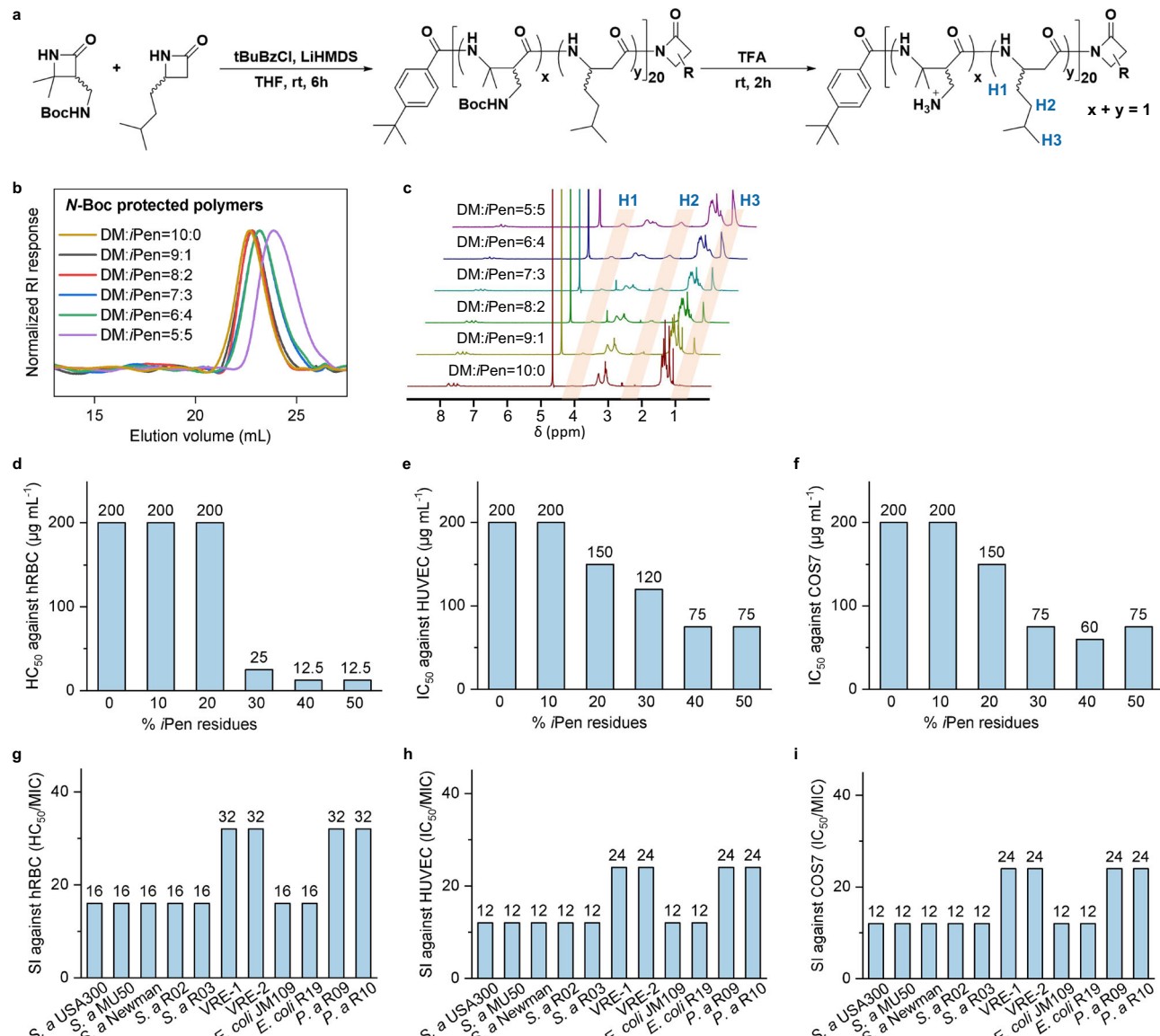

**Fig. 7 | Experimental validation. a** Synthesis of host defense peptides-mimicking $\beta$-amino acid polymers DM$_x$$i$Pen$_y$ (x + y = 1, y = 0–0.5), R represents the side chain from one of the starting monomers of DM and $i$Pen. **b** Gel permeation chromatography (GPC) traces of $N$-Boc-protected DM$_x$$i$Pen$_y$. **c** Proton nuclear magnetic resonance ($^1$H NMR) characterization of $N$-Boc-deprotected DM$_x$$i$Pen$_y$. H$_1$, H$_2$ and H$_3$ represent the characteristic peaks from $i$Pen component within polymers. **d** Hemolysis of polymers against human red blood cells (hRBCs). HC$_{50}$ means the

minimum concentration to cause 50% hemolysis. **e** Cytotoxicity of polymers against Human umbilical vein endothelial cell line (HUVEC) cells. IC$_{50}$ means the minimum concentration to cause 50% inhibition. **f** Cytotoxicity of polymers against African green monkey kidney fibroblasts (COS7) cells. **g–i** Selectivity index (SI) of the optimal polymer DM$_{0.8}$$i$Pen$_{0.2}$ calculated from HC$_{50}$/MIC against hRBCs, IC$_{50}$/MIC against HUVEC cells and COS7 cells. MIC means the minimum inhibitory concentration.

**Table 2 | Gel permeation chromatography (GPC) characterization of *N*-Boc protected DM$_x$iPen$_y$**

| x:y | $M_n$(g mol$^{-1}$) | DP | D |
|---|---|---|---|
| 10:0 | 5100 | 22 | 1.12 |
| 9:1 | 4900 | 22 | 1.13 |
| 8:2 | 5020 | 23 | 1.15 |
| 7:3 | 4500 | 22 | 1.13 |
| 6:4 | 4400 | 23 | 1.12 |
| 5:5 | 3600 | 20 | 1.09 |

$M_n$ means the obtained number average molecular weight, *D* means dispersity index, *DP* means degree of polymerization.

**Table 3 | Minimum inhibitory concentration (MIC) values of library DM$_x$iPen$_y$(x:y) against clinically isolated drug-resistant bacterial**

| Strain | MIC($\mu$g mL$^{-1}$) | | | | | |
|---|---|---|---|---|---|---|
| | 10:0 | 9:1 | 8:2 | 7:3 | 6:4 | 5:5 |
| *Staphylococcus aureus* USA300 | 25 | 12.5 | **12.5** | 12.5 | 12.5 | 12.5 |
| *Staphylococcus aureus* MU50 | 25 | 12.5 | **12.5** | 12.5 | 12.5 | 25 |
| *Staphylococcus aureus* Newman | 50 | 25 | **12.5** | 12.5 | 12.5 | 50 |
| *Staphylococcus aureus* RO2 | 25 | 12.5 | **12.5** | 12.5 | 12.5 | 25 |
| *Staphylococcus aureus* RO3 | 25 | 25 | **12.5** | 12.5 | 12.5 | 25 |
| *Vancomycin resistant enterococcus*-1 | 12.5 | 6.25 | **6.25** | 6.25 | 6.25 | 12.5 |
| *Vancomycin resistant enterococcus*-2 | 12.5 | 6.25 | **6.25** | 6.25 | 6.25 | 12.5 |
| *Escherichia coli* JM109 | 25 | 12.5 | **12.5** | 12.5 | 12.5 | 25 |
| *Escherichia coli* R19 | 50 | 25 | **12.5** | 12.5 | 12.5 | 25 |
| *Pseudomonas aeruginosa* R09 | 12.5 | 6.25 | **6.25** | 6.25 | 6.25 | 25 |
| *Pseudomonas aeruginosa* R10 | 12.5 | 12.5 | **6.25** | 6.25 | 6.25 | 25 |

Values in bold indicate the performance of DM$_{0.8}$iPen$_{0.2}$ which is chosen as the optimal antibacterial candidate.

that the candidate has selective antibacterial activity and potential application[11,67,68], proving the discovery of promising antimicrobial alternatives. It was worth noting that DM$_{0.8}$iPen$_{0.2}$ possessed a unique structure characterized by different hydrophobic subunits in comparison to previously reported antimicrobial $\beta$-amino acid polymers. Importantly, our prediction showed that DM$_{0.8}$iPen$_{0.2}$ had lower cytotoxicity and improved antimicrobial selectivity compared to amphiphilic polymers reported earlier, which utilized DM as the cationic subunit and hydrophobic subunits with lower carbon numbers, such as DM:CHx, DM:$\beta$CP, DM:$\beta$CH[42].

### Antimicrobial mechanism study of $\beta$-amino acid polymer (DM$_{0.8}$iPen$_{0.2}$)$_{20}$

We investigated the antimicrobial mechanisms of the optimal polymer (DM$_{0.8}$iPen$_{0.2}$)$_{20}$ against drug-resistant positive and drug-resistant negative bacteria. For the representative gram-positive bacteria of *S. aureus*, we conducted cytoplasmic membrane depolarization assay using DiSC3(5) dye as the bacterial membrane potential probe and cytoplasmic membrane permeability assay using propidium iodide (PI) dye as nucleic acid staining reagent to evaluate the interaction between (DM$_{0.8}$iPen$_{0.2}$)$_{20}$ and bacterial membrane. It was found that (DM$_{0.8}$iPen$_{0.2}$)$_{20}$ displayed a significant depolarization effect on *S. aureus* comparable to Triton X-100 (TX-100) and a strong membrane permeabilization effect (Fig. 8a, b). Scanning Electron Microscope (SEM) characterization demonstrated that the cell membrane of (DM$_{0.8}$iPen$_{0.2}$)$_{20}$ treated *S. aureus* have obvious damage compared to untreated and normal *S. aureus* (Fig. 8c). In addition, we conducted the time-laps fluorescent confocal imaging to observe a dynamic sterilization process using the green fluorescent dye-labeled (DM$_{0.8}$iPen$_{0.2}$)$_{20}$. After treating *S. aureus* with dye-labeled (DM$_{0.8}$iPen$_{0.2}$)$_{20}$ at $1 \times$MBC (minimum bactericidal concentration), it was observed that (DM$_{0.8}$iPen$_{0.2}$)$_{20}$ with green fluorescence and PI with red fluorescence entered into the bacteria cytoplasm almost simultaneously at about 30s, which echoed the strong membrane permeabilization effect (Fig. 8d). The above experiments all implied an antimicrobial mechanism by which (DM$_{0.8}$iPen$_{0.2}$)$_{20}$ killing drug-resistant *S. aureus* by strong interaction with bacteria membrane. For the representative gram-negative bacteria of *E. coli*, we found that (DM$_{0.8}$iPen$_{0.2}$)$_{20}$ have strong outer membrane perturbation ability via outer membrane permeabilization test (Fig. 8e). Continuous studies indicated that (DM$_{0.8}$iPen$_{0.2}$)$_{20}$ displayed a strong depolarization and permeabilization effect against *E. coli*, which was consistent with experimental results of wrinkles appearing on the membrane surface of (DM$_{0.8}$iPen$_{0.2}$)$_{20}$ treated *E. coli* in SEM characterization (Fig. 8f, g, Supplementary Fig. 56). Moreover, the confocal imaging of dynamic sterilization process demonstrated that (DM$_{0.8}$iPen$_{0.2}$)$_{20}$ with green fluorescence was gradually enriched on the membrane surface and then PI with red fluorescence started to entered into the bacteria cytoplasm (Fig. 8h). All those experimental results indicated that (DM$_{0.8}$iPen$_{0.2}$)$_{20}$ killed drug-resistant *E. coli* via antibacterial mechanism of membrane damage.

## Discussion

Artificial intelligence (AI) has already made significant contributions to the entire life-cycle of drug design. However, there is currently a lack of efficient AI methods specifically tailored for designing host defense peptide-mimicking polymers, mainly due to the scarcity number of available polymers in each family and multi-constraints when exploring the vast high-dimensional polymer space. In this study, we have developed an end-to-end AI-guided inverse design framework to realize effective exploration of novel host defense peptide-mimicking polymers under the conditions of 86 few-shot polymer data.

By applying multi-modal polymer representations, we extract multi-scale polymer information to improve the accuracy of the predictive model for few-shot data setting. All quantitative results prove a high reliability and stability of the predictive model, which can be further applied for the design process. Moreover, we distill the knowledge of our $\beta$-amino acids data and the natural $\alpha$-amino acids data, helping to construct a more concentrated chemical space for exploration. Thus, the generative model is able to efficiently generate polymers with high chemical rationality and synthetic feasibility under multiple constraints on desired bioactivities, toxicity and structures. Through iterative prediction and generation in reinforcement learning, we generate more than $10^5$ novel cationic-hydrophobic $\beta$-amino acid polymers, and we finally find 83 optimal polymers with the desired properties.

We also synthesize one of the predicted candidates, DM$_{0.8}$iPen$_{0.2}$, and verify the bioactivities. This polymer displays broad-spectrum and potent antibacterial activity and desirable antibacterial selectivity, indicating the effectiveness and feasibility of our AI strategy. Furthermore, our proposed data-driven AI strategy exhibits robust adaptability and holds great potential for application in various other domains beyond just a few-shot polymer or molecular systems. Through the utilization of our AI framework, we open up fresh opportunities to tackle the pressing challenge of efficiently identifying promising antibacterial polymers to counteract the growing threat of antibiotic resistance. In future studies, it worth exploring the AI-guided antimicrobial polymer design on more backbone types of polymers and more factors, such as various polymer descriptors, to more effectively find antimicrobial polymer candidates belonging to diverse species.

## Methods
### Data preprocessing
We made same data preprocessing to all the antibacterial activity data including MIC values as well as HC$_{10}$ values. If the end point of bacterial

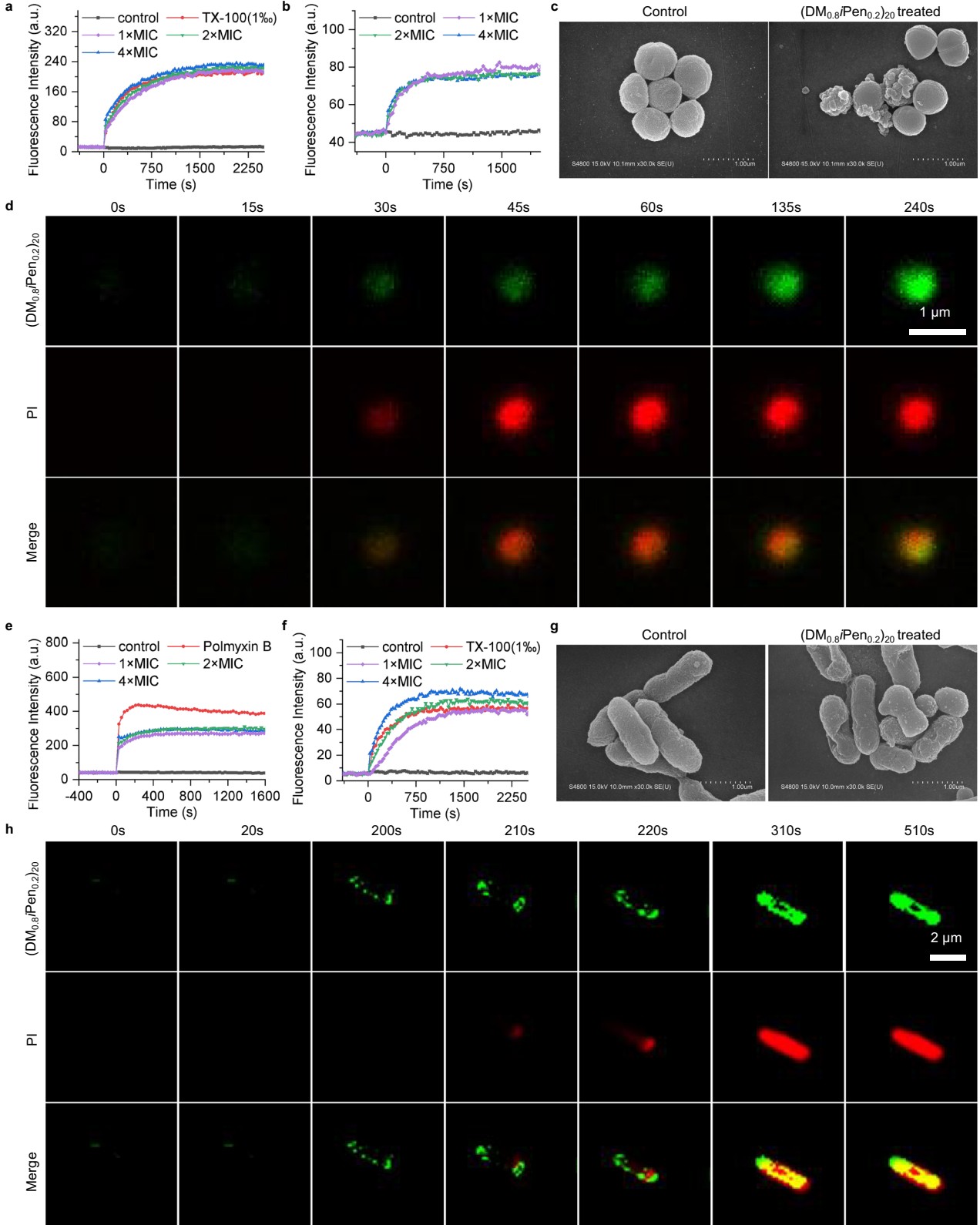

**Fig. 8 | Antimicrobial mechanism study. a** Cytoplasmic membrane depolarization of $(DM_{0.8}iPen_{0.2})_{20}$ against *S. aureus* USA300. TX-100 means Triton X-100. MIC means the minimum inhibitory concentration. **b** Cytoplasmic membrane permeability of $(DM_{0.8}iPen_{0.2})_{20}$ against *S. aureus* USA300. **c** Scanning Electron Microscope (SEM) characterization on *S. aureus* USA300 with and without $(DM_{0.8}iPen_{0.2})_{20}$ treatment at 1 × MBC. The SEM sample was prepared once, and at least 50 fungal cells were observed individually in the sample, showing results similar to the representative SEM images shown in the figure. MBC means minimum bactericidal concentration. **d** Time-laps confocal fluorescence imaging on the interaction between *S. aureus* USA300 and fluorescent dye-labeled $(DM_{0.8}iPen_{0.2})_{20}$ at 1 × MBC, in the presence of propidium iodide (PI). **e** Outer membrane permeability of $(DM_{0.8}iPen_{0.2})_{20}$ against *E. coli* R19. **f** Cytoplasmic membrane depolarization of $(DM_{0.8}iPen_{0.2})_{20}$ against *E. coli* R19. **g** SEM characterization on *E. coli* R19 with and without $(DM_{0.8}iPen_{0.2})_{20}$ treatment at 1 × MBC. The SEM sample was prepared once, and at least 50 fungal cells were observed individually in the sample, showing results similar to the representative SEM images shown in the figure. **h** Time-laps confocal fluorescence imaging on the interaction between *E. coli* R19 and fluorescent dye-labeled $(DM_{0.8}iPen_{0.2})_{20}$ at 1 × MBC, in the presence of PI.

growth had not been arrived during the experiment, the values was estimated to be the current available value. (e.g., for the experimental estimated value $HC_{10} > 400 \, \mu g \, mL^{-1}$, it was seen as $400 \, \mu g \, mL^{-1}$). Note that natural logarithm transformation was performed to all the estimated results due to the regular values so that all the results were transformed as integers labels for corresponding properties (e.g., "12.5" was recorded as "3", "400" was recorded as "8"). All models were trained to predict the natural logarithm of all properties.

## Polymer data augmentation

An important property for cationic-hydrophobic $\beta$-amino acid polymers, or more specifically for multi-component polymers is that the machine learning or deep learning model used should follow the permutation invariance of the polymer input, i.e. the results of the model should not be influenced by the order of the components, and it could be formulated as,

$$\begin{cases} H = M_1[(p_1,r_1),(p_2,r_2)] = M_1[(p_2,r_2),(p_1,r_1)], \\ S = M_2[(p_1,r_1),(p_2,r_2)] = M_2[(p_2,r_2),(p_1,r_1)], \\ E = M_3[(p_1,r_1),(p_2,r_2)] = M_3[(p_2,r_2),(p_1,r_1)], \end{cases} \quad (1)$$

where $M_1, M_2, M_3$ were different map functions from the polymer input to corresponding properties, $H, S, E$ were the value of $MIC_{S.aureus}$, $MIC_{E.coli}$ and $HC_{10}$, respectively, $p, r$ were the polymer unit and its composition information. In the previous work[47], it had been proved that by considering the permutation invariance, the model accuracy can be improved. In this way, we reasonably introduced this property as a method for data augmentation, aiming at improving the accuracy of the predictive model. Detailed, we adjusted the order of the input cationic and hydrophobic subunits and the feature orders in all representations were also changed with the same property label, so as to avoid the influence of the input order.

## Multi-modal random polymer representation

Translating polymers into machine readable vectors was one important problem with ongoing concerns for polymer informatics[69]. Different from micromolecules with deterministic topology connections of atoms and bonds, it was hard to completely represent random polymers with unregular sequence by general representation methods for micromolecules (e.g., SMILES or graphs) due to the intrinsically stochastic nature of polymers[44]. Generally considering, the property of a polymer was mainly decided by 1) structures of subunits and 2) subunit sequence connection, while for random polymers, the subunit ratio should be taken into consideration instead of sequence connection. In our work, we proposed a multi-modal polymer representation method from the following three perspectives:

**Molecular descriptors.** Molecular descriptors are mathematical representation of chemicals which are generally used to build predictive models. We used an open-sourced Mordred calculator[45], which included 1826 two- and three-dimensional descriptors. For cationic-hydrophobic polymers, descriptors of both the cationic and hydrophobic subunits were calculated and stacked together, totally dimensioned 3654 for candidate descriptor vector with adding composition information $r_1, r_2$ of two subunits. Then we applied a two-stage descriptor downselection strategy with a stage of statistical downselection and a stage of machine learning based downselection[46]. In the first stage, constant or almost constant descriptors were dropped from the initial set (Init., 3654 descriptors), and descriptors with variance larger than 10% of the mean value across the initial set were filtered out as validate set (Var., 1014 descriptors). Next, we evaluated Spearman rank correlations of each descriptor pair, and descriptors with correlation higher than 0.9 as well as correlation with the target property ($MIC_{S.aureus}$, $MIC_{E.coli}$ and $HC_{10}$) lower than 0.05 were filtered out as correlation set (Cor., 182, 171, 174 descriptors for $MIC_{S.aureus}$,

$MIC_{E.coli}$ and $HC_{10}$, respectively). In the second stage, a recursive feature elimination (RFE) method[70] was introduced on the Cor. descriptors set based on a random forest (RF) model. With RF regression, each descriptor was eliminated recursively according to the importance rankings until the last descriptor. Then, a 15-fold cross-validation was adopted with repeated stratified subsampling descriptors. The principle of choosing the optimized descriptor set was to choose a descriptor which has the lowest mean RMSE. In this way, descriptors with most important information related on the target property were selected. In our work, we chose 40 descriptors as the optimized molecular descriptors (Opt., 40 descriptors for $MIC_{S.aureus}$, $MIC_{E.coli}$ and $HC_{10}$, respectively) for part of the input of the predictive model (Results of selected descriptors are shown in Supplementary Figs. 3–8, and supplied predictive results are shown in Supplementary Fig. 9).

**Molecular representations.** Molecular representations are another popular ways to encode molecules. In recent polymer informatics, BigSMILES is a recently developed structurally-based line notation to reflect the stochastic nature of polymer molecules[44]. Compared with molecular descriptors, hidden chemical information could be learned from molecular representations via a data-driven pattern. According to the syntax of BigSMILES, we developed two kinds of other rules to completely define cationic-hydrophobic $\beta$-amino acid polymers, and also these rules are universal for other random polymers.

*Sequence representation.* Traditional SMILES strings generally consisted of various atom tokens (e.g., "C", "O", "[NH3+]"), bond tokens (e.g., "=", "#") and branching tokens (e.g., "()", "1,2") to encode molecules. In BigSMILES sequence, the stochastic object and the bonding descriptors were two new joined elements compared with basic SMILES grammar. We further introduced several additional definition so as the composition information of each repeated subunit in the stochastic object could be expressed, which was not included in BigSMILES. Take $DM_{0.6}BU_{0.4}$ as an example, it could be written as: {[>]NC(C)(C)C(C[NH3+])C=O.[+rn = 60], NC(CCCC)CC=O[<].[+rn = 40]}, where "[+rn = 60]" showed that the DM subunit has the ratio of 60%. ">" and "<" were two conjugate types of boding descriptors showing how repeat units were linked. For simplicity, we omitted exterior strings (since they are all same for our cationic-hydrophobic $\beta$-amino acid polymers) and we used the simplification style. Other cationic-hydrophobic polymers were defined like such. After collecting all characters involved, the one-hot encoding of the BigSMILES strings could be generated as the input. All the sequences are written manually and it is hard to be applied to large-scale datasets for further performance comparison, since there is still not mature toolkit for polymers.

*Graph representation.* Similarly, we construct graph representation for random polymers according to the BigSMILES syntax, shown in Supplementary Fig. 10. In BigSMILES syntax, bond descriptors are introduced to specify where and how repeat units can be joined with another repeat unit. Bonding descriptors are placed on atoms of a repeat unit that could form direct bonds with another repeat unit. In BigSMILES, there are two types of bonding descriptors: one is the "$" descriptor, or AA-type descriptor, which means it can only be connected with the same descriptors; the other is the "<" and ">" descriptors, or AB-type descriptor, which means one descriptor should be connected with the conjugate descriptor. These rules are translated into our tasks to represent a cationic-hydrophobic amphiphilic $\beta$-amino acid polymer.

## Predictive network

We testified the property prediction performance using various of representations and we set Morgan fingerprints, which was widely used in polymer property prediction[35], as baseline. In this study, we mainly used the following combinations according to three proposed multi-modal representations with properly designed network structures for specific tasks[71,72]: 1) Descriptor vector (from Descriptor_Init to

Descriptor_Opt), 2) Sequence vector, 3) Graph vector, 4) Sequence vector and Descriptor vector (Seq+Descriptor_Opt), 5) Graph vector and Descriptor vector (Graph+Descriptor_Opt), 6) Sequence vector, graph vector and Descriptor vector (Seq+Graph+Descriptor_Opt). Noted that we used the optimized descriptors for fusing since they had reached better model performance.

**Network architectures.** For situation 1), we transformed the descriptor feature $F_f$ by subtracting the means and dividing by the standard deviations as normalization process and we simply trained a Fully-connected Feed-forward Neural Network (FFN) for prediction. The dimensionality of the input feature $F_j$ is $[B, N_D]$ and the dimensionality of the input layer of FNN is $[N_D, D_f]$, where $B$ is the number of batch size, $N_D$ is the number of descriptors used and $D_f$ is the dimensionality of the hidden layers in FNN. For 2), we used the bidirectional Gate Recurrent Unit (GRU)[73,74] to extract the hidden information embedded in Sequence vector, and can be formulated as,

$$\begin{cases} \overrightarrow{h_k} = \overrightarrow{\mathbf{GRU}}(t_k, \overrightarrow{h_{k-1}}), \\ \overleftarrow{h_k} = \overleftarrow{\mathbf{GRU}}(t_k, \overleftarrow{h_{k-1}}), \end{cases} \tag{2}$$

where $t_k$ was the token embedding, and $\overrightarrow{h_k}, \overleftarrow{h_k}$ were bidirectional hidden states for the $k_{th}$ token of a string embedded by GRU, and the current hidden state $h_k$ was obtained as,

$$h_k = (\overrightarrow{h_k}, \overleftarrow{h_k}). \tag{3}$$

Finally, we used $F_s$ to denote the contextual representation of a sequence string with length $n$ as,

$$F_s = (h_0, h_1, \cdots, h_n). \tag{4}$$

The dimensionality of the input sequence vector is $[B, n]$ and the dimensionality of the sequence embedding is $[n, D_s]$, where $B$ is the number of batch size, $n$ is the number of each input sequence and $D_s$ is the dimensionality of the hidden layers in GRU. The final dimensionality of the sequence feature $F_s$ is $[B, D_s]$.

For 3), we apply a Bidirectional Message Communication GNN[75], which makes full use of the node message for more effective message interactions to extract the local information embedded in the graph. The network structures can be seen in Supplementary Fig. S11 and the pseudocode of the model were concluded in Supplementary Information as Algorithm 1.

Specifically, the input of the algorithm is each polymer graph $G=(\mathcal{V}, \mathcal{E})$ and all of its atom attributes $x_v(\forall v \in \mathcal{V})$ and bond attributes $x_{e_{vw}}(\forall e_{vw} \in \mathcal{E})$. The initial node feature $h_v^0$ is simply the atom attributes, while the initial edge feature $h_{e_{vw}}^0$ is the bond attributes. Then, according to the network depth $T$, a $T$ steps message aggregation and update procedure is applied. In each step $t$, each node message vector $m_v^{t+1}$ is aggregated according to its incoming edges and each edge message vector $m_{e_{vw}^{t+1}}$ is aggregated according to its neighbor nodes, shown as,

$$\begin{cases} m_v^{t+1} = \mathbf{MAX}(h_{e_{uv}}^t) \odot \mathbf{SUM}(h_{e_{uv}}^t), u \in \mathcal{N}(v), \\ m_{e_{vw}}^{t+1} = \mathbf{MEAN}(h_v^t, h_w^t), \end{cases} \tag{5}$$

where **MAX, SUM, MEAN** are the corresponding aggregating strategy, $\odot$ is an element-wise multiplication operator. Then the obtained message vectors of node and edge $m_v^{t+1}, m_{e_{vw}}^{t+1}$ are concatenated with the corresponding current hidden states to be sent to the communicate function which use an addition operator as communicative kernel to calculate the communicative vector $p_v^{t+1}, p_{e_{vw}}^{t+1}$. Then the hidden state of the node and edge are updated with skip connection as,

$$\begin{cases} h_v^{t+1} = U_v^t(p_v^{t+1}, h_v^0) = \mathbf{ReLU}(h_v^0 + \mathbf{W_v} \cdot p_v^{t+1}), \\ h_{e_{vw}}^{t+1} = U_e^t(p_{e_{vw}}^{t+1}, h_{e_{vw}}^0) = \mathbf{ReLU}(h_{e_{vw}}^0 + \mathbf{W_e} \cdot p_{e_{vw}}^{t+1}), \end{cases} \tag{6}$$

where **ReLU** is the rectified linear unit and $\mathbf{W_v}, \mathbf{W_e}$ are learned matrices. After $T$ step iteration, a GRU based readout function is applied to the final node representation $h_v^T$ to get the graph-level representation $F_g$ as,

$$F_g = \sum_{v \in \mathcal{V}} \mathbf{GRU}(h_v^T). \tag{7}$$

The dimensionality of the input atom vector and bond vector in graph are $[B, N_v, F_v]$ and $[B, N_e, F_e]$, and the dimensionality of the atom embedding and bond embedding in Bidirectional Message Communication GNN are $[F_v, D_g]$ and $[F_e, D_g]$, where $B$ is the number of batch size, $N_v, N_e$ are the atom number and bond number in each input molecular graph, $F_v$ and $F_e$ are the number of attributes for each atom and bond and $D_g$ is the dimensionality of the hidden layers in GNN. The final dimensionality of the graph feature $F_g$ is $[B, D_g]$. For 1)-3), the network structures can be seen in Supplementary Fig. 11.

Since 4), 5) and 6) involved multiple polymer vectors, we developed a multi-modal polymer representation method with adjustable network blocks for specific representations. A core motivation was how to learn more abundant chemical information from limited data points and how to find connections and differences between information in diverse representations. From feature descriptors, various basic chemical or calculated information could be gained. In contrast, from sequence or graph representations, distributions of atoms and bonds on spatial and numerical were explicitly displayed, while more implicit information, which might not be calculated through a specific equation, was generally learned with the help of data-driven deep learning. Since the available data are very limited, to learn better polymer feature for few-shot prediction, we tempted to merge various representations which is one of the main contributions of our work.

The main structure included several customized representation learning blocks to extract implicit information from various representations (descriptors, sequence and graph here), and this process could be formulated as,

$$F_j = \mathbf{Combine}(F_f, F_s, F_g), \tag{8}$$

where **Combine** was the function to assemble different representations with adding and stacking, and $F_j$ was the joint feature by stacking all the vector features with the dimensionality of $[B, D_j]$, $D_j = D + N_D$ ($D = D_f = D_s = D_g$). According to the different input representations in 4), 5) and 6), different blocks are inserted as shown in Supplementary Fig. 12.

Then a Transformer-based feature combination block was built with the input of $F_j$. The Transformer had been proved as a powerful model on various fields through its power on extracting comprehensive information. To find connections between the learned implicit information from Sequence and Graph representation and the explicit information embedded in descriptors, we further used descriptors $F_f$ as the attention bias in the self-attention mechanism, and this process could be formulated as:

$$Q = F_j W_Q, K = F_j W_K, V = F_j W_V, \tag{9}$$

$$\mathbf{Attention}(F_j) = \mathbf{softmax}(QK^\top / \sqrt{d_K} + F_f)V, \tag{10}$$

where $W_Q$, $W_K$, $W_V$ were the corresponding projecting matrices of $Q$ (query), $K$ (keys), $V$ (values), $d_K$ was the dimension of keys, **Attention** was the self-attention mechanism in Transformer and **softmax** was the softmax function. With the calculation of the Transformer block and feedforward network (FNN) block, we got the final predictions of the properties with the dimensionality of $[B,1]$,

$$P = \mathbf{FNN}(\mathbf{Transformer}(F_j, F_f)). \tag{11}$$

**Predictive model training settings.** We randomly split the training data $D_{train\_aug}$ into 8:1:1 train/valid/test ratios and we applied bayesian optimization to find the optimal hyper-parameters. Then we used the optimized parameters to retrain the model for 10 independent runs with different random seeds. Specifically, a dynamic changed learning rate was used with the Adam optimizer with mean squared error (MSE) loss to train the model. We set an initial learning rate as $10^{-4}$ and it would be doubled as a max learning rate in 5 warm up training epochs, and finally the learning would return the $10^{-4}$ as a final value. The training epoch and the batch size were set as 100 and 16 respectively. In each epoch, if the validation MSE reduced, the model would be saved. The parameters of each block of GNN, GRU, Transformer and FNN were all recorded in Supplementary Table 2. In addition, since the operation of random data splitting would cause uneven distribution of training data, we applied the ensembling technique, which is a common technique in machine learning. Multiple independently trained models with different random seeds were combined to produce an averaging predictions so as to prevent overfitting on partial results. After training, the unseen testing data $D_{test}$ was used to evaluate the performance of the model, using the R-squared coefficient (R2, higher R2 means better performance of the model) and root-mean-squared error (RMSE, lower RMSE means better performance of the model) as metrics. The implementation of the model relies on Pytorch and RDKit package.

**Scaffold-decorator generative network**
Take the hydrophobic subunit "BU" as an example, its SMILES string was "NC(CCCC)C=O", which could also be seen as that a side chain "[*]CCCC" was decorated to the scaffold "NC([*])CC=O", where "[*]" was the special attachment token for substitution. For scaffold with more than one substitution, a symbol "|" was introduced to differentiate decorations[76]. Therefore, the core problem of polymer design could be transformed as finding the optimized decoration for the specific scaffold to formulate subunits for polymer with desirable properties. We summarized the whole designing procedure in two stages. Firstly, we pre-trained a GRU-based molecular scaffold-decorator with the ability of generating valid subunits. Secondly, a reinforcement learning fine-tuning stage was adopted to explore the chemical space for optimal polymers. When fine-tuning, each reasonable molecule would be recorded for the convenience of final analysis and evaluation.

**Network architectures.** The implementation of scaffold-decorator network was totally an encoder-decoder architecture with attention mechanism. The encoder was a bidirectional RNN sequenced with an embedding layer and three layers of bidirectional GRU cells of 256 dimensions. Then the hidden states were sent to the decoder, which was a single direction RNN sequenced with an embedding layer, three layers of GRU cells of 256 dimensions. Finally, an global attention layer as adopted to sum up the output of the encoder and the decoder, and a liner layer was connected to calculate the probability of each possible token $x_i$. The model was trained to maximize the Negative Log-Likelihood (NLL) loss written as:

$$\mathrm{NLL}(\mathbf{S}) = -\sum_{i=1}^{n} \log P(x_i | x_{<i}, \mathrm{scaffold}), \tag{12}$$

where $P(x_i | x_{<i})$ was the probability when sampling the $i_{th}$ token of decoration sequence $\mathbf{S}$ with given the previous tokens and the input scaffold.

**Graph grammar distillation**
A direct idea was that the subunits for which we would like to explore had similar structures thus these structures must distributed closely in the huge chemical space (Supplementary Fig. 24). Generally, our $\beta$-amino acids have similar structures with natural $\alpha$-amino acids. However, if we pre-training our model with large-scale public data, those rules for constructing complex structures or undesirable chemical elements (e.g., Br,Cl) may also be embedded in the model. Thus, it takes a long time to further RL fine-tuning to adjust the parameters to avoid generating those subunits. Thereby, we first collected all cationic and hydrophobic $\beta$-amino acids in our data and several natural $\alpha$-amino acids structures (Supplementary Fig. 23). Then, we used a hyper-graph based data-efficient graph grammar learning method (DEG)[77] to collect various graph grammar rules from the given amino acids. Thus, various grammar rules were learned automatically from the training data, and specific rules could be learned according to the given data if needed. By doing so, we extracted grammar knowledge and we recombined these grammar to construct a distilled set of molecules. Then, we similarly used the RECAP rules to slice molecules, gaining 0.3 million pairs of scaffold-decoration data. We took these data to pre-train a generative model with the same structures above and constructed a more-focused chemical space embedded in the model for further exploration, so as to accelerate the search efficiency under multi constraints for RL agent.

**Reinforcement learning**
To further guide our generative model pre-trained by graph grammar distillation toward relevant areas in chemical space according to customized requirements, we adopted REINVENT 2.0[78]. It is a recently developed reinforcement learning method for de novo drug design, for fine-tuning to carry out a constellation of specific tasks of design. By fine-tuning, various user-defined requirements could be satisfied to generate molecules of interest. In our cases, we realized the following requirements: 1) polymer generation under various scaffold subunit structures (e.g., "NC([*])CC=O", "NCC(C=O)1[*]C1", "NC1[*]C1C=O"), 2) polymer generation under multi-objective constraints (e.g., carbon numbers, ring number and MIC$_{S.aureus}$/MIC$_{E.coli}$/HC$_{10}$ thresholds).

The main roles in REINVENT 2.0 included a prior model $M_{\mathrm{Prior}}$, an agent model $M_{\mathrm{Agent}}$ and a score modulating block. The prior model was the pre-trained scaffold-decorator generative model introduced above, while the agent model shared the identical network structures and the initialization parameters of the agent model as completely the same as the prior model. The score modulating block could be regarded as the environment which fed back rewards according to the targeted scoring functions.

Then we introduced the reinforcement learning cycle. First, the agent model $M_{\mathrm{agent}}$ sampled batch of SMILES decorations for a specific scaffold, and the decorated polymer were scored according to the scoring function $S_{\mathrm{score}}$ (introduced in Eq. (15)). Among each course of sampling molecules, the agent chose the next possible token, seen as the action $A_{\mathrm{action}}$, according to the current token sequence, regarded as the state $S_{\mathrm{state}}$ in the RL framework. Thus, the agent learned a conditional probability $p(A|S)$ to generate the desired molecules when the episodes go on. To train the agent model, we used the NLL, similar to Eq. (12), to represent the agent likelihood of the generated decoration sequence $\mathbf{S}$ as NLL$(\mathbf{S})_{\mathrm{Agent}}$. Then $\mathbf{S}$ would be given to the prior model $M_{\mathrm{Prior}}$ to calculate the augmented likelihood with the score $S_{\mathrm{score}}(\mathbf{S})$. Ultimately, the loss of the agent could be calculated as:

$$\mathrm{NLL}(\mathbf{S})_{\mathrm{Augmented}} = \mathrm{NLL}(\mathbf{S})_{\mathrm{Prior}} - \sigma S_{\mathrm{score}}(\mathbf{S}), \tag{13}$$

$$\text{loss} = [\text{NLL}(\mathbf{S})_{\text{Augmented}} - \text{NLL}(\mathbf{S})_{\text{Agent}}]^2, \tag{14}$$

where $\sigma$ was the scalar value to scale up the output of the score function. During the training process, we collected all valid generated molecules for data analysis, and molecules with desired properties would be further filtered out for experimental validation.

**Generative model training settings.** For the graph grammar distillation pre-training process, the training epoch, batch size and the learning rate were set as 450, 256 and $10^{-3}$ respectively. The dimensionality of hidden layers of GRU was set as 256. For the reinforcement learning fine-tuning process, the training epoch, batch size and the learning rate were set as 450, 30 and $10^{-9}$ respectively. Also, we use the negative log likelihood (NLL) loss to train the model and the implementation of the model relies on Pytorch and RDKit package. All hyperparameters are concluded in Supplementary Table 5.

**Score and metric**
In this study, we aimed to find more potential cationic and hydrophobic subunit combinations with specific composition rations, which satisfied the desired properties: $\text{MIC}_{S.aureus} < 25$, $\text{MIC}_{E.coli} > 25$ and $\text{HC}_{10} > 100$. Moreover, we designed several penalty rules or customized constraints to accelerate the learning process with narrowing down the scope of exploration. The final score function $S_{\text{score}}(\mathbf{S})$, which could also be seen as the reward in RL, was written as:

$$S_{\textbf{score}}(\mathbf{S}) = S_{\text{property}}(\mathbf{S}) + S_{\text{penalty}}(\mathbf{S}) + S_{\text{constrain}}(\mathbf{S}), \tag{15}$$

where $S_{\text{property}}$, $S_{\text{penalty}}$, $S_{\text{constrain}}$ were three different parts of target activities, penalty of irrationality and customlized constraints for scoring.

**Property score.** We used the previously trained predictive network to calculate the value of $\text{MIC}_{S.aureus}$, $\text{MIC}_{E.coli}$ and $\text{HC}_{10}$ respectively. For all the calculated values, we applied score transformations (sigmoid for $\text{HC}_{10}$ and reverse sigmoid for $\text{MIC}_{S.aureus}$ and $\text{MIC}_{E.coli}$) so that each component returned a value between [0,1] (the higher the better). This operation helped to avoid one-sided impacts of single-properties and adjust the influence of multi-parameter objectives, and the property scoring function could be written as:

$$S_{\text{property}}(\mathbf{S}) = a * \text{MIC}_{S.aureus}(\mathbf{S}) + b * \text{MIC}_{E.coli}(\mathbf{S}) + c * \text{HC}_{10}(\mathbf{S}), \tag{16}$$

where $a$, $b$ and $c$ were adjustable weights showing that which the agent should put more focus on. They were decided by customized design demands. In this work, we focused more on the antimicrobial activity and hence, we set a larger value for $a = b = 2$ than $c = 1$.

**Penalty score.** To improve the rationality and correctness of the generated molecules, we designed several structural penalties as the penalty scoring function,

$$S_{\text{penalty}}(\mathbf{S}) = \begin{cases} -5, & \text{when molecule is invalid}, \\ -3, & \text{when unexpected elements exist}. \end{cases} \tag{17}$$

**Constrain score.** To further constrain the structures of the generated molecules, we also designed several constraints which can be used alternatively,

$$S_{\text{constrain}}(\mathbf{S}) = \begin{cases} C/2, & C_{\text{number}} \leq X, \\ -2, & C_{\text{number}} > X, \end{cases} + \begin{cases} -3, & R > Y, \\ 0, & R \leq Y, \end{cases} \tag{18}$$

where $C_{\text{number}}$ was the carbon number of the final decorated molecule, $R$ was the ring number and $X$, $Y$ were adjustable constants to decide how many carbon atoms or how many circles should be generated. As discussed before, to prove the rationality of the generated polymers, we set $X = 11$ and $Y = 1$ in our exploration settings to prove the toxicity without chemical structural rationality.

**Evaluation settings**
To exactly find new candidate polymers with desired properties, we set three situations to evaluate the performance of the generative model pre-trained by graph grammar distillation. It is worth noting that in all situations, we fixed the cationic monomer as DM and MM structures and focused on designing new hydrophobic monomers. This helped to improve the synthetic possibility for final validation. Even new cationic monomers were not taken into consideration. It was still a challenging problem since there still existed multi-constraints (structures, properties etc.) and should be taken into consideration to adjust hydrophobic subunits. Details are outlined below: **Task 1**: Cationic: DM/MM, hydrophobic: any scaffold, reward design:

$$S_{\text{score}}(\mathbf{S}) = 2 * \text{MIC}_{S.aureus}(\mathbf{S}) + S_{\text{penalty}}(\mathbf{S}). \tag{19}$$

**Task 2**: Cationic: DM/MM, hydrophobic: any scaffold, reward design:

$$\begin{aligned} S_{\text{score}}(\mathbf{S}) = &2 * \text{MIC}_{S.aureus}(\mathbf{S}) + 2 * \text{MIC}_{E.coli}(\mathbf{S}) \\ &+ 1 * \text{HC}_{10}(\mathbf{S}) + S_{\text{penalty}}(\mathbf{S}) + S_{\text{constrain}}(\mathbf{S}), \end{aligned} \tag{20}$$

where $X = 11$, $Y = 1$. **Task 3**: Cationic: DM, hydrophobic: NC([*])C C(=O), reward design:

$$\begin{aligned} S_{\text{score}}(\mathbf{S}) = &1 * \text{MIC}_{S.aureus}(\mathbf{S}) + 1 * \text{MIC}_{E.coli}(\mathbf{S}) + 3 * \text{HC}_{10}(\mathbf{S}) \\ &+ S_{\text{penalty}}(\mathbf{S}) + S_{\text{constrain}}(\mathbf{S}), \end{aligned} \tag{21}$$

where $X = 11$, $Y = 1$.

Task 1 was designed to evaluate the performance of the model pre-trained by graph grammar distillation and the model pre-trained by ChEMBL. Task 2 referred to analysis the distributions of the generated polymers with evaluations by the predictive model, aiming at further find the desirable area with more possible optimal candidates under multiple-constrains. Following the results in Task 2, we made further qualifications on a specific scaffold structure, aiming exploring more candidate polymer with specific scaffold for real-work synthesis, as Task 3. Note that we defined a high weight for $\text{HC}_{10}$ which was mainly due to the fact that for polymers with a fixed DM subunit, most polymers showed undesirable property on $\text{HC}_{10}$ and we gave more weights on it.

**Materials**
All chemical reagents and solvents were used without further purification. Anhydrous dichloromethane (DCM) and anhydrous Tetrahydrofuran (THF) were purchased from Sigma-Aldrich. Ethyl acetate (EtOAc) and other solvents were purchased from Shanghai Titan Technology Co., Ltd. Synthesized chemicals were purified using a SepaBean machine equipped with Sepaflash columns produced by Santai Technologies Inc in China. The water used in these experiments was obtained from a Millipore water purification system with a resistivity of 18.2 MΩ.cm. 3-(4,5-Dimethylthiazol-2-yl)-2,5-diphenyltetrazolium bromide (MTT) was purchased from MACKLIN regent, Shanghai. Dulbecco's modified Eagle medium (DMEM) were purchased from Hyclone (USA).

## Cell lines

Human umbilical vein endothelial cell line (HUVEC) and the African green monkey kidney fibroblasts (COS7) cells were obtained from the Cell Bank of the Chinese Academy of Sciences (Shanghai, China).

## Measurements

Nuclear magnetic resonance (NMR) spectra were collected on a Bruker spectrometer at 400 MHz using CDCl$_3$ as the solvent and 600 MHz using D$_2$O as the solvent. The corresponding chemical shifts were referenced to residual protons in the deuterated NMR solvents. High resolution electrospray ionization time-of-flight mass spectrometry (HRESI-MS) was collected on a Waters XEVO G2 TOF mass spectrometer. Gel permeation chromatography (GPC) was performed on a Waters GPC instrument equipped with a refractive index detector (Waters 2414) using dimethylformamide (DMF), supplemented with 0.01 M LiBr, as the mobile phase at a flow rate of 1 mL min$^{-1}$ at 50 °C. The GPC were equipped by a Tosoh TSKgel Alpha-2500 column (particle size 7 μm) and a Tosoh TSKgel Alpha-3000 column (particle size 7 μm) linked in series. Relative number-average molecular weight ($M_n$), degree of polymerization (DP) and dispersity index (D) were calculated from a calibration curve using polymethylmethacrylate (PMMA) as standards. Before GPC characterization, all samples were filtered through 0.22 μm polytetrafluoroethylene (PTFE) filters. Optical density (OD) value and fluorescence value were recorded on a multifunction microplate reader (SpectraMax M2).

## Synthesis of β-lactams

(±)-3-tert-Butyloxycarbonylaminomethyl-4,4-dimethyl azetidin-2-one (3, β-lactam DM, Supplementary Fig. 57) was synthesized by following previously reported procedure[43]. Briefly, 3,3-Dimethylallyl bromide (15.0 g, 0.1 mol) and potassium phthalimide (20.4 g, 0.11 mol) potassium phthalimide was mixed in 300 mL DMF. The reaction mixture was stirred vigorously at room temperature for 16 h and then poured into 800 mL ice water with vigorous stirring to result precipitate. The precipitate was collected by filtration and washing with ethanol to give the crude product. After removing the solvent under vacuum, the intermediate compound 1 (Supplementary Fig. 57) was directly used without purification (20.0 g, 93.0%). $^1$H NMR (400 MHz, CDCl$_3$, Supplementary Fig. 58): δ 7.90-7.78 (m, 2H), 7.72-7.66 (m, 2H), 5.27 (dt, J = 7.2, 1.2 Hz, 1H), 4.5 (d, J = 7.2 Hz, 2H), 1.83 (s, 3H), 1.70 (s, 3H). HRESI-MS (Supplementary Fig. 59): m/z calculated for C$_{13}$H$_{14}$NO$_2$ [M+H]$^+$: 216.1025; Found: 216.1024.

To a solution of the intermediate compound 1 (20.0 g, 0.093 mol) in dichloromethane (50 mL) was added chlorosulfonyl isocyanate (10.4 mL, 0.11 mol) under N$_2$ atmosphere. The reaction mixture was stirred for 30 min at 0 °C and then warmed up to room temperature for 72 h. Then the reaction mixture was poured into a suspension of Na$_2$SO$_3$ (41.6 g, 0.33 mol) and Na$_2$HPO$_4$ (46.9 g, 0.33 mol) in water (800 mL) and was stirred for 12 h. The aqueous phase was extracted with dichloromethane (3 × 500 mL). The organic phase was combined and then dried over anhydrous magnesium sulfate and concentrated under vacuum to give the crude product. The crude product was purified by recrystallization from ethyl acetate and hexane to afford maleimide-protected DM (compound 2, Supplementary Fig. 57) as white solid (14.6 g, 60.8%). $^1$H NMR (400 MHz, CDCl$_3$, Supplementary Fig. 60): δ 7.89-7.81 (m, 2H), 7.77-7.67 (m,2H), 6.00 (br, 1H), 4.09 (dd, J = 8.0, 14.0 Hz, 1H), 3.91 (dd, J = 8.0, 14.0 Hz, 1H), 3.41 (t, J = 8.0 Hz, 1H), 1.47 (s, 3H), 1.45 (s, 3H). HRESI-MS (Supplementary Fig. 61): m/z calculated for C$_{14}$H$_{15}$N$_2$O$_3$ [M+H]$^+$: 259.1083; Found: 259.1084.

To a solution of above compound 2 (14.6 g, 56.5 mmol) in methanol (200 mL) was added a solution of hydrazin hydrate (80% solution in water, 14 mL). The reaction mixture was stirred at 70 °C for 12 h to result precipitate. After removing the precipitate by

filtration, the filtrate was coevaporated with toluene (3 × 200 mL) for removing the residual hydrazine hydrate. The residue, di-tert-butyl dicarbonate (Boc$_2$O, 24.6 g, 113.0 mmol) and triethylamine (15.3 mL, 113.0 mmol) were mixed in 500 mL methanol. The reaction mixture was refluxed for 6 h. After filtration, the solvent was concentrated under vacuum to give the residue. The residue was dissolved in dichloromethane (200 mL) and then washed sequentially with hydrochloric acid solution (1 N), sodium hydroxide solution (1 N) and brine (200 mL). The organic phase was dried over anhydrous magnesium sulfate and then concentrated under vacuum to give the crude product, which was directly purified by column chromatography to obtain β-Lactam DM (compound 3, Supplementary Fig. 57) as white solid (9.7 g, 75.2%). $^1$H NMR (400 MHz, CDCl$_3$, Supplementary Fig. 62): δ 5.9 (s, 1H), 4.9 (s, 1H), 3.66-3.56 (m, 1H), 3.28 (t, J = 10.2 Hz, 1H), 2.97 (t, J = 7.8 Hz, 1H), 1.45 (s, 3H), 1.43 (s, 9H), 1.37 (s, 3H). $^{13}$C NMR (100 MHz, CDCl$_3$, Supplementary Fig. 63): δ 169.01, 155.79, 79.58, 58.24, 54.80, 37.07, 28.62, 28.37, 22.86. HRESI-MS (Supplementary Fig. 64): m/z calculated for C$_{11}$H$_{20}$N$_2$NaO$_3$ [M+Na]$^+$: 251.1372; Found: 251.1371.

4-(2-methylpropyl)azetidin-2-one (compound 4, β-lactam iPen, Supplementary Fig. 57) was synthesized according to the method reported in previous literature[66]. Briefly, to a solution of 5-Methyl-1-hexene (3.5 g, 35.6 mmol, 1.0 equiv.) in dichloromethane (10 mL), chlorosulfonyl isocyanate (3.3 mL, 37.4 mmol, 1.05 equiv.) was added at 0 °C under N$_2$ atmosphere. The reaction mixture was stirred for 3 days at room temperature then monitored by thin layer chromatography (TLC). The reaction was quenched via carefully transferring into the buffer (200 mL) consisting of anhydrous sodium sulfite (13.5 g, 106.8 mmol, 3.0 equiv.) and disodium hydrogen phosphate (15.2 g, 106.8 mmol, 3.0 equiv.), the mixture was stirred for overnight and extracted with dichloromethane (3 × 100 mL), then the organic layer was combined and dried over anhydrous MgSO$_4$. After removing the solvent under vacuum, the crude product was purified by silica gel column chromatography to afford β-lactam iPen (compound 4) as colorless oil (2.1 g, 41.7% yield). $^1$H NMR (400 MHz, CDCl$_3$, Supplementary Fig. 65): δ 6.06 (s, 1H), 3.60-3.54 (m, 1H), 3.04 (ddd, J = 14.8, 4.8, 2 Hz, 1H), 3.04 (dq, J = 14.8, 1.2 Hz, 1H), 1.69-1.49 (m, 3H), 1.26-1.10 (m, 2H), 0.89 (d, J = 6.8 Hz, 6H). $^{13}$C NMR (100 MHz, CDCl$_3$, Supplementary Fig. 66): δ 168.87, 48.22, 43.09, 35.07, 33.14, 27.70, 22.34. HRESI-MS (Supplementary Fig. 67): m/z calculated for C$_8$H$_{15}$NNaO [M+Na]$^+$:164.1051; Found 164.1053.

## Synthesis of β-amino acid polymers

All polymerizations of β-lactams were carried in nitrogen-regulated glove box at room temperature. Initiator (tBuBzCl), β-lactams (DM and iPen) and the base catalyst LiHMDS were dissolved in dry THF to a concentration of 0.2 M respectively. The positive charge and hydrophobic composition of β-amino acid polymers was controlled by the initial feed ratio of tBuBzCl: DM: iPen. Briefly, 2 mL β-lactams with different volume ratios of DM: iPen were mixed, after adding 100 μL tBuBzCl solution in THF and 300 μL LiHMDS in THF into the mixture sequentially, the reaction mixture was stirred for 12 h. When the polymerization reaction was completed, the reaction mixture was poured into cold petroleum ether (PE, 45 mL) to precipitate out the crude product as a white solid, followed by centrifugation (2810 g) to remove the solvent. The crude product was dissolved in 2 mL THF followed by pouring cold petroleum ether (45 mL) to precipitate out the crude product. The N-Boc protected polymers were purified by dissolution-precipitation process using THF/PE (2 mL/45 mL) three times and vacuum drying for overnight to give a white solid. The number-average molecular weight ($M_n$) and polydispersity index (D) were characterized by GPC using N, N-dimethylformamide (DMF) as the mobile phase.

N-Boc protected polymers were dissolved in trifluoroacetic acid (2 mL). Then the mixture was under shaking for 2 h at room

temperature. After removing the solvent under a nitrogen flow, the residue was dissolved in methanol (1 mL), followed by addition of cold MTBE (45 mL) to precipitate out the crude polymers. The crude polymers were purified by three times of dissolution-precipitation process using methanol/MTBE (1 mL/45 mL) and vacuum drying for overnight. The purified polymers were dissolved in Milli-Q water and lyophilized to obtain a white powder in the form of TFA salt (>80.0% yield), which was further characterized by $^1$H NMR and used for antibacterial assay.

## Minimum inhibitory concentration (MIC) assay

11 strains of gram positive and negative bacteria were respectively cultured in Luria-Bertani (LB) medium for 9 h at 37 °C under shaking at 200 rpm. After centrifugation at 4000 rpm for 5 min, the bacteria in the culture medium were collected and re-suspended in Mueller-Hinton (MH) medium to $2 \times 10^5$ CFU mL$^{-1}$ as the working suspension. The deprotected polymer DM$_x$iPen$_y$ libraries were respectively diluted to concentrations ranging from 3.13 μg mL$^{-1}$ to 400 μg mL$^{-1}$ by a two-fold gradient dilution in a 96-well plate. After mixing equal volumes of bacterial cells suspension (50 μL) and DMxiPeny solution (50 μL) in each well, the 96-well plates were incubated at 37 °C for 9 h to collect OD values on a SpectraMax®M2 plate reader. MH medium was used as the blank; bacteria in MH medium was used as positive control. The percentage of bacterial cells survival was calculated from the equation below:

$$\text{Cell growth (\%)} = \frac{OD_{600}^{polymer} - OD_{600}^{blank}}{OD_{600}^{control} - OD_{600}^{blank}} \times 100. \tag{22}$$

The MIC value was defined as the lowest concentration of an antimicrobial agent to completely inhibit microbial growth. Measurements were performed in duplicates, and the experiments were repeated at least twice.

## Hemolysis assay

Fresh human blood was washed with Tris-buffered saline (TBS, pH = 7.2) for three times and the collected human red blood cells (hRBCs) were diluted to 5% (v/v) with TBS to obtain working suspension. DM$_x$iPen$_y$ solution were diluted to concentrations ranging from 3.13 to 400 μg mL$^{-1}$ by a two-fold gradient dilution in a 96-well plate. After mixing an equal volume of hRBCs suspension and DM$_x$iPen$_y$ solution, 96-well plates were incubated at 37 °C for 1 h. TBS was used as the blank, the mixture of Triton X-100 (0.1% in TBS) and hRBCs was used as the positive control. After centrifugation, 80 μL of the supernatant in each well was transferred to another 96-well plate and the optical density (OD) value was collected at 405 nm. The percentage of hemolysis was calculated from

$$\text{hemolysis (\%)} = \frac{OD_{405}^{polymer} - OD_{405}^{blank}}{OD_{405}^{control} - OD_{405}^{blank}} \times 100. \tag{23}$$

The HC$_{50}$ was defined as the concentration of a compound to cause 50% hemolysis. Measurements were performed in triplicate. The experiments were repeated three times independently.

All sourced blood for hemolysis assays were donated by the Shanghai RuiJin Rehabilitation Hospital before the blood is disposed as scheduled and no recruitment information was supplied to the researchers of this project as per the agreement with University of East China University of Science and Technology Human Ethics Approval, therefore recruitment information are unknown.

## Cytotoxicity assay

The cytotoxicity of $\beta$-amino acid polymers was studied using MTT assay. Specifically, COS-7 and HUVEC cells were respectively incubated in Dulbecco's Modified Eagle's Medium (DMEM) containing FBS (10%) and penicillin/streptomycin (1%) at 37 °C in a humidified atmosphere containing 5 °C CO$_2$. Cells were seeded in a 96 well plates at 5000 cells in 100 μL DMEM medium for each well and the plates were incubated at 37 °C in a humidified atmosphere containing 5% CO$_2$ for 24 h. Different concentrations of DM$_x$iPen$_y$ solution ranging from 400 μg mL$^{-1}$ to 3.13 μg mL$^{-1}$ were prepared and added to HUVEC and COS-7 cells, respectively. The plates were incubated for another 48 hours. An aliquot of 10 μL MTT solution (5 mg mL$^{-1}$) in phosphate buffered saline (PBS) was added in each well and the plate was incubated for 4 h. After removing the supernatant, 150 μL DMSO was added in each well and then the plate was shaken for 15 min before measuring the absorbance at 570 nm on a microplate reader. The untreated cells were used as positive control, DMEM solution was used as blank. The percentage of cell viability was calculated from

$$\text{Cell viability (\%)} = \frac{A_{570}^{polymer} - A_{570}^{blank}}{A_{570}^{control} - A_{570}^{blank}} \times 100. \tag{24}$$

The IC$_{50}$ was defined as the minimum concentration to cause 50% inhibition. All experiments were carried out with three replicates. Each experiment was repeated at least twice.

## Synthesis of dye-labeled (DM$_{0.8}$iPen$_{0.2}$)$_{20}$

The dye-labeled $\beta$-amino acid polymer (DM$_{0.8}$iPen$_{0.2}$)$_{20}$ was synthesized according to the protocol in our previous study[79]. Briefly, the initiator Dye-NHS ester, co-initiator LiHMDS, DM monomer and iPen monomer were dissolved in dried tetrahydrofuran (THF) to the solution with a final concentration of 0.2 M inside a glove box. DM (1.6 mL), iPen (0.4 mL) and Dye-NHS ester (0.1 mL) were mixed and stirred. Subsequently, LiHMDS (0.3 mL) was quickly added into the mixture. The reaction mixture was stirred for 6 h at room temperature and then quenched with 5 drops of MeOH. After removing the solvent under N$_2$ flow, the residue was dissolved in THF (1 mL) and transferred to a centrifuge tube, followed by slowly addition of cold PE (45 mL) into the mixture to precipitate out a yellow product. The N-Boc protected polymer was further purified via three times of dissolution-precipitation process using the solvent of THF/PE (1 mL/45 mL) then vacuum dried and characterized by GPC using DMF containing 10 mM LiBr as mobile phase. This polymer was dissolved in 2 mL TFA and stirred at room temperature for 2 h to remove the Boc protection. After removing the TFA under N$_2$ flow, the residue was dissolved in MeOH (1 mL), followed by slow addition of cold methyl tert-butyl ether (MTBE, 45 mL) to precipitate out a yellow product. The N-Boc deprotected polymer was further purified via three times of dissolution-precipitation process using the solvent of MeOH/MTBE (1 mL/45 mL) then vacuum dried and dissolved in Milli-Q water. Subsequently, the solution was subjected to lyophilization to give a final dye-labeled polymer (DM$_{0.8}$iPen$_{0.2}$)$_{20}$, which was used for confocal imaging.

## Time-lapse fluorescent confocal imaging assay

The confocal imaging assay for drug-resistant S. aureus and drug-resistant E. coli was conducted according to the protocol in our previous study[80]. Briefly, the dye-labeled (DM$_{0.8}$iPen$_{0.2}$)$_{20}$ (4 × MBC, green fluorescence) and propidium iodide (40 mM, red fluorescence) were mixed in equal volumes to prepare a working solution. In addition, the bacteria were cultured in LB medium at 37 °C for 6 h to obtain the bacterial suspension, the bacterial suspension was washed by PBS buffer and then diluted in MH medium to achieve a working suspension with a cell density of $1 \times 10^7$ CFU mL$^{-1}$. 10 μL of the bacterial suspension was dropped into a glass-bottomed cell culture dish for 10 min to allow the bacteria to attach to the bottom. Subsequently, 10 μL of working solution was added to the bacterial drop. The confocal images were captured at the various time points for three channels: bright field, 488 nm (green fluorescence) and 562 nm (red fluorescence),

respectively. These images were used to record the bactericidal process.

## Outer membrane permeabilization assay

The outer membrane permeabilization assay for drug-resistant *E. coli* was conducted according to the protocol in our previous study[43]. Briefly, the bacteria were cultured in LB medium at 37 °C for 6 h to obtain the bacterial suspension, the bacterial suspension was washed by PBS buffer and then diluted in HEPES medium (5 mM HEPES, 20 mM glucose, pH = 7.4) to achieve a working suspension with a cell density of $3 \times 10^8$ CFU mL$^{-1}$, followed by addition of 1-N-phenyl-naphthylamine (NPN) dye at a final concentration of 10 μM. 90 μL of working suspension containing NPN was added to each well of a 384-well plate. The fluorescence changes (excitation $\lambda = 350$ nm, emission $\lambda = 420$ nm) were recorded on a SpectraMax®M2 plate reader (Molecular Devics, USA). Once the fluorescence intensity remained stable, 10 μL of (DM$_{0.8}$*i*Pen$_{0.2}$)$_{20}$ was added to the bacterial solution, and the fluorescence intensity was recorded continuously.

## Cytoplasmic membrane depolarization assay

The cytoplasmic membrane depolarization for drug-resistant *S. aureus* and drug-resistant *E. coli* was conducted according to the protocol in our previous study[43]. The drug-resistant bacteria were cultured in LB medium at 37 °C for 6 h, and then the bacterial suspension was diluted in HEPES medium (5 mM HEPES, 20 mM glucose, pH = 7.4) to achieve a working suspension with a cell density of $1 \times 10^7$ CFU mL$^{-1}$, followed by addition of 3, 3'-dipropylthiadicarbocyanine iodide (diSC3(5)) dye at a final concentration of 0.8 μM. The bacterial suspension was incubated for 1 h, followed by the addition of KCl to a final concentration of 0.1 M to balance the cytoplasmic and external K$^+$ concentration. 90 μL of bacterial suspension containing diSC3(5) was added to each well of a 384-well plate. The fluorescence changes (excitation $\lambda = 622$ nm, emission $\lambda = 673$ nm) were recorded on a SpectraMax®M2 plate reader (Molecular Devics, USA). Once the fluorescence intensity remained stable, 10 μL of (DM$_{0.8}$*i*Pen$_{0.2}$)$_{20}$ and 0.1% Triton X-100 as the positive control was separately added to the bacterial solution and the fluorescence intensity was recorded continuously.

## Cytoplasmic membrane permeabilization assay

The cytoplasmic membrane permeabilization assay for drug-resistant *S. aureus* and drug-resistant *E. coli* was conducted according to the protocol in our previous study[43]. The drug-resistant bacteria were cultured in LB medium at 37 °C for 6 h, and then the bacterial suspension was diluted in HEPES medium (5 mM HEPES, 5 mM glucose, pH = 7.4) to achieve a working suspension with a cell density of $1 \times 10^8$ CFU mL$^{-1}$, followed by addition of propidium iodide (PI) dye at a final concentration of 10 μM. 150 μL of bacterial suspension containing PI was added to each well of a corning 96-well plate. The fluorescence changes (excitation $\lambda = 535$ nm, emission $\lambda = 617$ nm) were recorded on a SpectraMax®M2 plate reader (Molecular Devics, USA). Once the fluorescence intensity remained stable, 10 μL of (DM$_{0.8}$*i*Pen$_{0.2}$)$_{20}$ was added to the bacterial solution and the fluorescence intensity was recorded continuously.

## SEM characterization of bacteria morphology

The SEM characterization for drug-resistant *S. aureus* and drug-resistant *E. coli* was conducted according to the protocol in our previous study[79]. Briefly, the drug-resistant bacteria were cultured in LB medium at 37 °C for 9 h, and then the bacterial suspension was diluted in LB medium to achieve a working suspension with a cell density of $1 \times 10^7$ CFU mL$^{-1}$, followed by addition of (DM$_{0.8}$*i*Pen$_{0.2}$)$_{20}$ at a final concentration of $1 \times$ MBC. The bacterial suspension was incubated at 37 °C for 30 min. An untreated bacteria suspension was used as the control. (DM$_{0.8}$*i*Pen$_{0.2}$)$_{20}$ treated and untreated bacteria were collected by centrifugation at 1700 × *g* for 5 min. They were washed with phosphate buffer saline (PBS) once and then fixed with 4% glutaraldehyde in phosphate buffer (PB) at 25 °C overnight. The bacteria were further washed with PBS and dehydrated with gradient ethanol (EtOH) solutions (30, 50, 70, 80, 90, 95, and then 100% ethanol). The samples were dried in air and then used for Field Emission Scanning Electron Microscopy (FESEM) characterization.

## Reporting summary

Further information on research design is available in the Nature Portfolio Reporting Summary linked to this article.

## Data availability

All collected raw data of *β*-amino acid polymers, *α*-amino acid polymers, polymethacrylates, polymethacrylamides and other categories are available on GitHub: https://github.com/TianyuWu813/polymer_prediction. All raw data to train the generative model are available on https://github.com/TianyuWu813/polymer_generation. The source data for all figures and tables in the manuscript and in the Supplementary Information are provided with this paper. Source data are provided with this paper.

## Code availability

Codes supporting this study are available on GitHub and Zenodo: https://github.com/TianyuWu813/polymer_prediction for polymer property prediction[81] and https://github.com/TianyuWu813/polymer_generation for polymer generation[82].

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

## Acknowledgements

This research was supported by National Natural Science Foundation of China (Basic Science Center Program: No. 61988101), National Natural Science Foundation of China (No. T2325010, No. 22075078, No.62233005 and No. 62293502), National Key Research and Development Program of China (2022YFC2303100), German Research Foundation DFG (Project No.411803875), Fundamental Research Funds for the Central Universities (222202417006), Shanghai Frontiers Science Center of Optogenetic Techniques for Cell Metabolism (Shanghai Municipal Education Commission), and the Programme of Introducing Talents of Discipline to Universities (the 111 Project) under Grant B17017 and Shanghai AI Lab. We also thank the Research Center of Analysis and Test of East China University of Science and Technology for the help with the characterization. The authors also thank the support of the Analysis and Testing Center of School of Chemical Engineering, East China university of Science and Technology. Thanks for the staff members of the Integrated Laser Microscopy System at the National Facility for Protein Science in Shanghai (NFPS), Zhangjiang Lab, China, for providing technical support and assistance in data collection and analysis. Please refer to Journal-level guidance for any specific requirements.

## Author contributions

Y.T. and R.L. directed the whole project and conceived the idea. T.W. and M.Z. designed the experiments and wrote the manuscript together. T.W. constructed the polymer inverse design framework. M.Z. contributed to the polymer synthesis,antibacterial mechanism study and the data analysis. F.Q and J.K contributed to result discussion. J.Z. conducted the antibacterial assay. Q.C. conducted the cytotoxic assay. All authors proofread the manuscript.

## Competing interests

The authors declare no competing interests.
