## [Peer Review File · Nature Communications]

AI-guided Few-shot Inverse Design of HDP-Mimicking Polymers against Drug-Resistant BacteriaREVIEWER COMMENTS

Reviewer #2 (Remarks to the Author):

The authors introduce a novel approach for predicting and generating HDP-mimicking polymers, especially under limited data scenarios. The predictive model incorporates diverse polymer representations, including sequence, graph, and physical descriptors, to enhance the richness of information. A grammar knowledge distillation technique, aimed at constraining the vast-dimensional polymer space, is employed for the pre-training of a generative model. Subsequently, the two models are seamlessly integrated using the Reinforcement Learning (RL) methodology to establish a comprehensive framework for polymer synthesis. It's noteworthy that the design of the model was backed by extensive comparative experiments.

The transformation between domain-specific knowledge and data representation in specific scenarios is eloquently addressed, showcasing a comprehensive consideration. The model design is notably thorough and well-conceived from an academic standpoint.

However, I have the following major concerns.

(1) Given the limited data volume and the similarity in scaffolds, a rigorous test set division should be employed to evaluate the proposed methodology. It would also be beneficial to consider incorporating polymers of other scaffolds for a more comprehensive assessment .

(2) There exist antibacterial polymers that do not fall under the category of β -amino acid polymers. It would be prudent for you to include predictions on these, as it would serve to validate the model's transferability and broaden its applicability.

(3)The synthesized polymer has rather typical MIC values, and the performance is average. It would be beneficial to supplement with experiments elucidating the antibacterial mechanism.

(4) The synthesized polymer requires further structural characterization.

(5) I would recommend the authors to explore and utilize additional machine learning models in the first stage, such as XGBOOST, to undertake this task for potentially enhanced results.

(6) In Fig. 2 , the best result in a does not align with those in d, and b exhibits the same issue.

(7) Equation (5) should be elaborated upon in greater detail, and within the "predictive network" section, it is essential to explicitly specify the dimensional changes of the variables within the model.

(8) The authors should elucidate the novelty in structure of the polymers selected for experimentation in comparison to previously discovered materials.

(9) Given the limited amount of data, the authors should provide a detailed description of the training process for the predictive model, as well as the methodology employed to select the optimal model to prevent overfitting

Reviewer #4 (Remarks to the Author):

The accurate prediction of antimicrobial polymers composed of unnatural amino acids is a long-term challenge due to limited polymer datasets. The manuscript describes the striking advance in accurate prediction and discovering on HDP-mimicking β -amino acid polymers via a few-shot polymer inverse design framework. The authors successfully identify 83 optimal antibacterial polymers via simulating prediction of over 105 polymers. This is a very interesting and important

paper in the field of research on the discovery of antibacterial compounds against drug-resistant bacteria. In addition, the author synthesized an optimal polymer and found that the polymer exhibited broad-spectrum and potent antibacterial activity against multiple clinically isolated antibiotic-resistant pathogens, validating the effectiveness of AI-guided design strategy. In summary, the reviewer suggests the acceptance of the article for publication in Nature Communications after addressing the minor points.

1. It is recommended to add the description of the importance of research on antimicrobial β -amino acid polymers and provide references to the latest literature.

2. Mass spectrometric characterization of positively charged monomer DM is suggested to be supplemented.

3. Both x , y and $x\%$, $y\%$ are used to present the ratio of the two subunits of β -amino acid polymers. The ratio of positively charged subunits to hydrophobic subunits should be standardized.

4. There is an error in the table of Fig 5, the table formatting should be corrected and the abbreviations should be given full descriptions.

5. The definition of the end group R in the chemical structure of antibacterial β -amino acid polymers in Fig 6 should be supplemented.

6. The author should check the writing standardization and labeling of chemical structures of polymers throughout the manuscript, such as heterochiral structures (\pm), superscripts in ^{13}C NMR, etc.

April 8th, 2024

Reply to Reviewers:

We are submitting the revision of our manuscript entitled “AI-guided Few-shot Inverse Design of HDP-Mimicking Polymers against Drug-Resistant Bacteria (Manuscript ID: NCOMMS-23-45033)” which we believe to be suitable for publication as a research article in *Nature Communications*.

We greatly thank the suggestive comments from reviewers. The suggestions for modification are very helpful, and the resulting changes have greatly improved the manuscript, in our view. Our responses to specific comments are detailed below; for reference, the comments are presented in *italics*, with our response following in blue color text.

We also made changes in the main text accordingly and kept all the changes in the revised manuscript highlighted using yellow color. We also uploaded a revised manuscript without highlighted changes.

We hope that the revised manuscript will prove to be acceptable for publication in *Nature Communications*.

Thank you for your great help with our manuscript.

Reviewer 1:

Comments:

The authors introduce a novel approach for predicting and generating HDP-mimicking polymers, especially under limited data scenarios. The predictive model incorporates diverse polymer representations, including sequence, graph, and physical descriptors, to enhance the richness of information. A grammar knowledge distillation technique, aimed at constraining the vast-dimensional polymer space, is employed for the pre-training of a generative model. Subsequently, the two models are seamlessly integrated using the Reinforcement Learning (RL) methodology to establish a comprehensive framework for polymer synthesis. It's noteworthy that the design of the model was backed by extensive comparative experiments. The transformation between domain-specific knowledge and data representation in specific scenarios is eloquently addressed, showcasing a comprehensive consideration. The model design is notably thorough and well-conceived from an academic standpoint.

Response: We thank the reviewer for the high recognition and positive comments. We also carefully revised our manuscript according to the reviewer's comments below.

1. However, I have the following major concerns. Given the limited data volume and the similarity in scaffolds, a rigorous test set division should be employed to evaluate the proposed methodology. It would also be beneficial to consider incorporating polymers of other scaffolds for a more comprehensive assessment.

Response: We thank the reviewer for the suggestion and we make responses in two parts, including the evaluation on a more rigorously divided test set and the assessment on the candidate polymer discovery with poly(α -amino acid) and polypeptoid scaffolds

1) Following the reviewer's suggestion, we redivided the test data according to the positively charged subunit and hydrophobic subunit for bioactivity prediction. We also summarized all the results with radar plot so as to clearly embody the difference between the predicted values and real measured values and the differences between different composition pairs (new Fig. 3a). Results indicated that our predictive model was capable of making credible predictions of the bioactivity of β -amino acid polymers. All added revisions are shown as below.

"We further compared in detail on bioactivity of all polymers between the predictive values and the real measured ones (Fig. 3a). We divided the data according to the positively charged subunit and hydrophobic subunit, so as to more rigorously embody the differences of the compositions (Fig. S1). Note that log transformation was performed to all the estimated results. The final R2 scores of our model reached 0.91, 0.88 and 0.91 on MIC_{S. aureus}, MIC_{E. coli} and HC₁₀ for DM series polymers, and 0.92, 0.84 and 0.96 on MM series polymers. It was obviously found from the radar plot that the predicted values highly fit real measured values, indicating that our predictive model was capable of making credible predictions of the bioactivity of β -amino acid polymers."

Fig. 3 a) Comparison between the predicted values and the real measured values. All results show that

our model can reach a desirable accuracy, with the metric of R2 reaching 0.91, 0.88 and 0.91 on $MIC_{S.aureus}$, $MIC_{E.coli}$ and HC_{10} for DM series polymers, and 0.92, 0.84 and 0.96 on MM series polymers. Both the predicted and real values are the transformed values by natural logarithm.

2) We further applied our model on several other scaffolds, including poly(α -amino acid) and polypeptoid scaffolds for more comprehensive assessment so as to expand the application of our model. The experimentally setting were almost the same to the main text. All results are concluded in supplementary information (new Fig. S51-S55). All added revisions are shown as below.

“In addition, we expanded our model on poly(α -amino acid) and polypeptoid scaffolds to explore further potential application of our method (Fig. S51-S55). All experimental settings were same, and we also screened out several broad-spectrum antibacterial candidate polymers.”

Fig. S51 Polymer discovery with α -peptide scaffold. a) Various hydrophobic α -amino acids generated in the discovery process with fixed D,L-Lysine cationic subunit. b) Property distribution of the predicted value of $MIC_{S.aureus}$, $MIC_{E.coli}$ and HC_{10} is shown. The units for all properties are ($\mu\text{g/mL}$).

Fig. S52 Polymer discovery with α -peptide/peptoid hybrid scaffold with N -substituted glycine peptoid. a) Various hydrophobic N -substituted glycine subunits generated in the discovery process with fixed D,L-Lysine cationic subunit. b) Property distribution of the predicted value of $MIC_{S.aureus}$, $MIC_{E.coli}$ and HC_{10} is shown. The units for all properties are ($\mu\text{g/mL}$).

Fig. S53 Polymer discovery with α -peptide/peptoid hybrid scaffold with disubstituted peptoid. a) Various hydrophobic disubstituted subunit generated in the discovery process with fixed D,L-Lysine cationic subunit. b) Property distribution of the predicted value of $MIC_{S.aureus}$, $MIC_{E.coli}$ and HC_{10} is shown. The units for all properties are ($\mu\text{g/mL}$).

Candidate	Structure	x:y	HC_{10} ($\mu\text{g/mL}$)	$MIC_{S.aureus}$ ($\mu\text{g/mL}$)	$MIC_{E.Coli}$ ($\mu\text{g/mL}$)
Candidate_α_1		2:8	186.57	28.84	33.08
Candidate_α_2		2:8	208.74	31.87	31.76
Candidate_α_3		5:5	101.81	18.19	20.67
Candidate_α_4		6:4	103.47	20.67	29.85

Fig. S54 α -amino acid subunit optimized structures. The units for all properties are ($\mu\text{g/mL}$). We default n to 20 for prediction.

Candidate	Structure	x:y	HC ₁₀ (μg/mL)	MIC _{S. aureus} (μg/mL)	MIC _{E. coli} (μg/mL)
Candidate_α_5		2:8	107.15	36.31	40.43
Candidate_α_6		6:4	207.10	21.49	37.84
Candidate_α_7		7:3	242.58	19.60	37.65
Candidate_α_8		7:3	137.27	19.31	36.80
Candidate_α_9		8:2	157.65	17.41	36.91

Fig. S55 α -amino acid subunit optimized structures. The units for all properties are ($\mu\text{g/mL}$). We default n to 20 for prediction.

- There exist antibacterial polymers that do not fall under the category of β -amino acid polymers. It would be prudent for you to include predictions on these, as it would serve to validate the model's transferability and broaden its applicability.

Response: We thank the reviewer for the concerns. Accurate prediction of antimicrobial activity and toxicity of polymers is especially challenging due to the random sequence of these polymers compared to sequence defined antimicrobial peptides as reported in recent AI-guided antimicrobial α -peptide design (*Nat. Biomed. Eng.* **2023**, 7, 797). Nevertheless, we take the challenge to explore the prediction on polymers bearing other backbones.

Following the review's suggestions, we reviewed a large amount of literature to collect antimicrobial polymers out of β -amino acid polymers. We classified them into several categories based on backbone structure: α -amino acid polymers (*Nat. Commun.* **2018**, 9, 5297), polymethacrylates (*J. Am. Chem. Soc.* **2005**, 127, 4128; *Polymers*, **2011**, 3, 1512; *Chem. Eur. J.* **2009**, 15, 1123; *ACS Macro Lett.*, **2014**, 3, 319), polymethacrylamides (*Biomacromolecules*, **2009**, 10, 3098) and polynorbornenes (*J. Am. Chem. Soc.*, **2004**, 126, 15870), pyridinium polymers (*Angew. Chem. Int. Ed.*, **2008**, 120, 1270) and poly(vinyl ether)s (*Biomacromolecules*, **2011**, 12, 3581). These polymers were collected for making predictions with our trained model to evaluate the transferability and applicability.

In the revised manuscript, we made predictions with our trained model and we visualized the results in new Fig. 3b-3d, exemplified with each one from α -amino acid polymers, polymethacrylates and polymethacrylamides to show our model's transferability and applicability (Fig. 3b-3d). Note that we use the metric of mean absolute error (MAE) to show direct difference of the transferability performance of our model in categories of antibacterial polymers. For α -amino acid polymers, the MAE was 0.51 and 0.79 for MIC_{S.aureus} and MIC_{E.coli}, which was nearly twice than the MAE for β -amino acid polymers (0.17 and 0.40 for MIC_{S.aureus} and MIC_{E.coli}) in a relatively small value (Fig. 3b). This fact suggested promising prospects for transferring our method to other categories of

antibacterial polymers that possess similar structural characteristics to β -amino acid polymers. For polymethacrylates, the MAE reaching 1.24 and 1.95 (nearly six times than β -amino acid polymers) for $MIC_{S.aureus}$ and $MIC_{E.coli}$, respectively (Fig. 3c). For polymethacrylamides, the MAE reached 2.33 and 3.75 (nearly over ten times than β -amino acid polymers) for $MIC_{S.aureus}$ and $MIC_{E.coli}$, respectively (Fig. 3d). These results showed that our model encountered challenges when predicting the properties of other polymers for example polymethacrylates and polymethacrylamides due to substantial dissimilarities with β -amino acid polymers. In summary, our model demonstrated promising transferability to α -amino acid polymers due to their similarity with β -amino acid polymers, while it encountered difficulties in transferring to scaffolds without similarities. In addition, we supplied complete predictive results in supplementary materials, shown in new Fig. S14-S22. All related added descriptions in revised manuscript are shown as below.

“Moreover, considering the variegation of antibacterial polymers and the rarity of partial types of polymers, we evaluated the transferability of our proposed method in order to broaden its applicability. We collected additional data on α -amino acid polymers[52], polymethacrylates [53-56], polymethacrylamides [57] and other categories [58-60] to evaluate the transferability of our model. Note that we use the metric of mean absolute error (MAE) to show direct difference of the transferability performance of our model in different categories of antibacterial polymers. According to the evaluated results, for α -amino acid polymers, the MAE was 0.51 and 0.79 for $MIC_{S.aureus}$ and $MIC_{E.coli}$, which was close to MAE of β -amino acid polymers (0.17 and 0.40 for $MIC_{S.aureus}$ and $MIC_{E.coli}$, Fig. 3b-3e). This fact suggested promising prospects for transferring our method to other categories of antibacterial polymers that possess similar structural characteristics to β -amino acid polymers. For polymethacrylates, the MAE reaching 1.24 and 1.95 (nearly six times than β -amino acid polymers) for $MIC_{S.aureus}$ and $MIC_{E.coli}$, respectively (Fig. 3f-3i). For polymethacrylamides, the MAE reached 2.33 and 3.75 (nearly over ten times than β -amino acid polymers) for $MIC_{S.aureus}$ and $MIC_{E.coli}$, respectively (Fig. 3j-3m). These results showed that our model encountered challenges when predicting the properties of other polymers for example polymethacrylates and polymethacrylamides due to substantial dissimilarities with β -amino acid polymers. In summary, our model demonstrated promising transferability to α -amino acid polymers which had highly similarity with our trained data of β -amino acid polymers, while our model were not suggested to be directly transferred to other categories before we further improve the performance of the model (All results shown Fig. S14-S22).”

We also supply additional descriptions in the discussion part about several future directions to improve the application of our method.

“In future studies, it worth exploring the AI-guided antimicrobial polymer design on more backbone types of polymers and more factors, such as various polymer descriptors, to more effectively find antimicrobial polymer candidates belonging to diverse species.”

Fig. 2 b-m) Comparison between the predicted values and the real measured values for α -amino acid polymers (b), polymethacrylates (c) and polymethacrylamides (d). Predicted values on $\text{MIC}_{S. aureus}$ (c, g, k) and $\text{MIC}_{E. coli}$ (d, h, l) in various proportion were recorded on the bar plot. Both the predicted and real values are the transformed values by natural logarithm. In addition, we visualized the difference on calculated MAE (e, i, m) for each polymer to show the difference when model transferring.

Poly(α -amino acids)	x:y	Predicted values ($\mu\text{g/mL}$)			Real values ($\mu\text{g/mL}$) ^[1]		
		HC ₅₀	$\text{MIC}_{S. aureus}$	$\text{MIC}_{E. coli}$	HC ₅₀	$\text{MIC}_{S. aureus}$	$\text{MIC}_{E. coli}$
	9:1		20.99	37.84		12	12
	8:2		16.03	35.52		25	12
	7:3		22.46	35.10		25	25
	6:4	N/A	19.46	37.07	N/A	25	25
	5:5		23.83	37.69		25	25
	4:6		24.92	43.74		50	50

Fig. S14 Quantitative predictive results for Poly(α -amino acid)s compared with real values. Values are reported from the corresponding literature^[1]. We default n to 20 for prediction. N/A is not available, meaning that real values are not reported in the literature and we do not make corresponding predictions.

Polymethacrylates	x:y	Predicted values (µg/mL)			Real values (µg/mL) ^{[2][3]}		
		HC ₅₀	MIC _{S.aureus}	MIC _{E.coli}	HC ₅₀	MIC _{S.aureus}	MIC _{E.coli}
	2:8	10.38		92.39	40		10
	2.8:7.2	10.81		92.08	16		1.1
	3.7:6.3	10.24	N/A	94.00	16	N/A	0.3
	4.5:5.5	9.62		95.69	16		0.8
	5.3:4.7	9.65		97.42	16		1
	0:10	132.94	61.72	923.63	>2000	125	500
	1.2:8.8	165.29	63.95	483.30	>2000	125	500
	2.8:7.2	173.47	56.04	557.51	>2000	250	500
	4.7:6.3	152.30	65.53	585.97	>2000	125	63
	6.3:3.7	180.02	57.18	680.04	114	125	16

Fig. S15 Quantitative predictive results for polymethacrylates compared with real values. Values are reported from the corresponding literature^[2,3]. We default n to 20 for prediction. N/A is not available, meaning that real values are not reported in the literature and we do not make corresponding predictions.

Polymethacrylates	x:y	Predicted values (µg/mL)			Real values (µg/mL) ^[4]		
		HC ₅₀	MIC _{S.aureus}	MIC _{E.coli}	HC ₅₀	MIC _{S.aureus}	MIC _{E.coli}
	8.8:1.2	N/A		258.87	N/A		460
	3:7	N/A		204.67	N/A		460
	5.6:4.4	N/A		206.30	N/A	N/A	13
	4.4:5.6	11.27	N/A	204.10	100		10
	4:6	11.40		205.47	15		10
	3.6:6.4	12.02		202.78	10		10
	9:1	N/A		188.49	N/A		190
	8.4:1.6	N/A		178.60	N/A		170
	6.5:3.5	N/A	N/A	158.50	N/A	N/A	16
	5:5	43.89		150.86	1.3		2.9
	3:7	45.38		148.32	0.15		7.7
	8.3:1.7	9.49		157.15	108		25
	7.6:2.4	9.22		144.49	15		13
	7.1:2.9	9.58		147.91	12		11
	6.8:3.2	9.77		138.37	6		8.8
	6.3:3.7	N/A	N/A	132.74	N/A	N/A	10
	6:4	9.13		128.40	1.7		5.3
	5.9:4.1	N/A		131.35	N/A		10
	5.6:4.4	9.23		123.82	1.3		10
	5.3:4.7	9.66		124.06	1.3		10

Fig. S16 Quantitative predictive results for polymethacrylates compared with real values. Values are reported from the corresponding literature^[4]. We default n to 20 for prediction. N/A is not available, meaning that real values are not reported in the literature and we do not make corresponding predictions.

Polymethacrylates	x:y	Predicted values (µg/mL)			Real values (µg/mL) ^[4]		
		HC ₅₀	MIC _{S.aureus}	MIC _{E.coli}	HC ₅₀	MIC _{S.aureus}	MIC _{E.coli}
	9.2:0.8	6.58		158.39	77		40
	8.4:1.6	6.40		146.25	10		12
	7.8:2.2	6.94	N/A	147.89	2.8	N/A	12
	7.3:2.7	6.92		141.90	1.3		12
	7.3	6.65		137.16	1		6
	9.2:0.8	1.91		88.27	175		29
	8.4:1.6	1.88		81.00	10		15
	7.5:2.5	2.01		76.00	1		4.6
	6.7:3.3	2.02	N/A	68.97	0.24	N/A	4.6
	5.6:4.4	1.97		N/A	0.17		N/A
	4.9:5.1	2.03		N/A	0.16		N/A

Fig. S17 Quantitative predictive results for polymethacrylates compared with real values. Values are reported from the corresponding literature^[4]. We default n to 20 for prediction. N/A is not available, meaning that real values are not reported in the literature and we do not make corresponding predictions.

Polymethacrylates	x:y	Predicted values (µg/mL)			Real values (µg/mL) ^[5]		
		HC ₅₀	MIC _{S.aureus}	MIC _{E.coli}	HC ₅₀	MIC _{S.aureus}	MIC _{E.coli}
	3.1:6.9		1.89			1500	
	1.9:8.1	N/A	1.31	N/A	N/A	94	N/A
	0.5:9.5		1.09			94	
	3.3:6.7		0.17			188	
	1.9:8.2	N/A	0.17	N/A	N/A	94	N/A
	0.6:9.4		0.17			47	

Fig. S18 Quantitative predictive results for polymethacrylates compared with real values. Values are reported from the corresponding literature^[5]. We default n to 20 for prediction. N/A is not available, meaning that real values are not reported in the literature and we do not make corresponding predictions.

Polymethacrylamides	x:y	Predicted values (µg/mL)			Real values (µg/mL) ^[6]		
		HC ₅₀	MIC _{S.aureus}	MIC _{E.coli}	HC ₅₀	MIC _{S.aureus}	MIC _{E.coli}
	0:10	197.00	56.80	345.34	>1000	1.7	19
	2:8	163.77	58.50	307.74	>1000	13	278
	3.6:6.4	131.23	59.66	290.37	>1000	100	464
	5.4:4.6	93.47	60.37	303.43	>1000	117	170
	7.8:2.2	71.96	64.17	356.89	>1000	46	100
	1.8:8.2	22.93	61.94	273.34	300	10	273.34
	3.3:6.7	20.59	56.11	257.12	12.5	13	257.12
	5.1:4.9	15.72	67.02	266.89	<0.16	11	266.89
	6.3:3.7	14.20	50.55	280.01	<0.16	13	280.01

Fig. S19 Quantitative predictive results for polymethacrylamides compared with real values. Values are reported from the corresponding literature^[6]. We default n to 20 for prediction. N/A is not available, meaning that real values are not reported in the literature and we do not make corresponding predictions.

Polynorbornenes	x:y	Predicted values (µg/mL)			Real values (µg/mL) ^[7]		
		HC ₅₀	MIC _{S.aureus}	MIC _{E.coli}	HC ₅₀	MIC _{S.aureus}	MIC _{E.coli}
	9:1	0.26		5897	>4000		40
	6.7:3.3	0.24	N/A	5855	>4000	N/A	40
	3.3:6.7	0.21		6017	<1		40
	2:8	0.2		6148	<1		40

Fig. S20 Quantitative predictive results for polynorbornenes compared with real values. Values are reported from the corresponding literature^[7]. We default n to 20 for prediction. N/A is not available, meaning that real values are not reported in the literature and we do not make corresponding predictions.

Pyridinium polymers	m	Predicted values (µg/mL)			Real values (µg/mL) ^[8]		
		HC ₅₀	MIC _{S.aureus}	MIC _{E.coli}	HC ₅₀	MIC _{S.aureus}	MIC _{E.coli}
	2	15.10		247.73	2393		600
	3	19.28		262.08	1897		200
	4	25.82		226.94	1709		30
	6	18.55	N/A	217.29	351	N/A	100
	8	17.89		215.47	229		450
	10	15.07		217.14	393		1100
	2	67.70		284.15	1147		350
	3	11.91		257.17	108		100
	4	5.08		247.71	0.23		15
	6	1.76	N/A	225.81	0.15	N/A	50
	8	1.23		221.29	0.11		125
	10	0.53		206.77	0.83		650

Fig. S21 Quantitative predictive results for pyridinium polymers compared with real values. Values are reported from the corresponding literature^[8]. We default n to 20 for prediction. N/A is not available, meaning that real values are not reported in the literature and we do not make corresponding predictions. The column for “m” means the number of atoms of carbon connected in the place.

Poly(vinyl ether)s	x:y	Predicted values (µg/mL)			Real values (µg/mL) ^[9]		
		HC ₅₀	MIC _{S.aureus}	MIC _{E.coli}	HC ₅₀	MIC _{S.aureus}	MIC _{E.coli}
	7.5:2.5	922.76		409.38	0.49		1.6
	4.7:5.3	894.94	N/A	380.92	1.8	N/A	3.1
	2.1:7.9	1062.14		375.50	18.9		31.3

Fig. S22 Quantitative predictive results for poly(vinyl ether)s compared with real values. Values are reported from the corresponding literature^[9]. We default n to 20 for prediction. N/A is not available, meaning that real values are not reported in the literature and we do not make corresponding predictions.

- The synthesized polymer has rather typical MIC values, and the performance is average. It would be beneficial to supplement with experiments elucidating the antibacterial mechanism.

Response: We thank the reviewer for the valuable comments and we have supplemented the antibacterial mechanism of the optimal polymer (DM_{0.8}iPen_{0.2})₂₀ against drug-resistant gram-positive and gram-negative bacteria to improve the manuscript. We also added the corresponding description on antibacterial mechanism in the revised manuscript as shown below (new Fig. 8).

“Antimicrobial mechanism study of β -amino acid polymer (DM_{0.8i}Pen_{0.2})₂₀. We investigated the antimicrobial mechanisms of the optimal polymer (DM_{0.8i}Pen_{0.2})₂₀ against drug-resistant Gram-positive and drug-resistant Gram-negative bacteria. For the representative gram-positive bacteria *S. aureus*, we conducted cytoplasmic membrane depolarization assay using DiSC3(5) dye as the bacterial membrane potential probe and cytoplasmic membrane permeability assay using propidium iodide (PI) dye as nucleic acid staining reagent to evaluate the interaction between (DM_{0.8i}Pen_{0.2})₂₀ and bacterial membrane. It was found that (DM_{0.8i}Pen_{0.2})₂₀ displayed a significant depolarization effect on *S. aureus* comparable to TX-100 and a strong membrane permeabilization effect (Fig. 8a-8b). Scanning Electron Microscope (SEM) characterization demonstrated that the cell membrane of (DM_{0.8i}Pen_{0.2})₂₀ treated *S. aureus* have obvious damage compared to untreated and normal *S. aureus* (Fig. 8c). In addition, we conducted the time-laps fluorescent confocal imaging to observe a dynamic sterilization process using the green fluorescent dye-labelled (DM_{0.8i}Pen_{0.2})₂₀. After treating *S. aureus* with dye-labelled (DM_{0.8i}Pen_{0.2})₂₀ at $1 \times$ MBC concentration, it was observed that (DM_{0.8i}Pen_{0.2})₂₀ with green fluorescence and PI with red fluorescence entered into the bacteria cytoplasm almost simultaneously at about 30 s, which echoed the strong membrane permeabilization effect (Fig. 8d). The above experiments all implied an antimicrobial mechanism by which (DM_{0.8i}Pen_{0.2})₂₀ killing drug-resistant *S. aureus* through strong interaction with bacteria membrane. For the representative gram-negative bacteria of *E. coli*, it was found that (DM_{0.8i}Pen_{0.2})₂₀ had strong outer membrane perturbation ability via outer membrane permeabilization test (Fig. 8e). Continuous studies indicated that (DM_{0.8i}Pen_{0.2})₂₀ displayed a strong depolarization and permeabilization effect against *E. coli*, which was consistent with experimental results of wrinkles appearing on the membrane surface of (DM_{0.8i}Pen_{0.2})₂₀ treated *E. coli* in SEM characterization (Fig. 8f-8g, Fig S56). Moreover, the confocal imaging of dynamic sterilization process demonstrated that (DM_{0.8i}Pen_{0.2})₂₀ with green fluorescence was gradually enriched on the membrane surface and then PI with red fluorescence started to enter into the bacteria cytoplasm (Fig. 8h). All those experimental results indicated that (DM_{0.8i}Pen_{0.2})₂₀ killed drug-resistant *E. coli* via antibacterial mechanism of membrane damage.”

In addition, we supply corresponding experimental details about the antimicrobial mechanism study in Methods shown as below.

“Synthesis of dye-labelled (DM_{0.8i}Pen_{0.2})₂₀. The dye-labelled β -amino acid polymer (DM_{0.8i}Pen_{0.2})₂₀ was synthesized according to the protocol in our previous study [78]. Briefly, the initiator Dye-NHS ester, co-initiator LiHMDS, DM monomer and *i*Pen monomer were dissolved in dried tetrahydrofuran (THF) to the solution with a final concentration of 0.2 M inside a glove box. DM (1.6 mL), *i*Pen (0.4 mL) and Dye-NHS ester (0.1 mL) were mixed and stirred. Subsequently, LiHMDS (0.3 mL) was quickly added into the mixture. The reaction mixture was stirred for 6 hours at room temperature and then quenched with 5 drops of MeOH. After removing the solvent under N₂ flow, the residue was dissolved in THF (1 mL) and transferred to a centrifuge tube, followed by slowly addition of cold PE (45 mL) into the mixture to precipitate out a yellow product. The N-Boc protected polymer was further purified via three times of dissolution-precipitation process using the solvent of THF/PE (1 mL/45 mL) then vacuum dried and characterized by GPC using DMF containing 10 mM LiBr as mobile phase. This polymer was dissolved in 2 mL TFA and stirred at room temperature for 2 h to remove the Boc protection. After removing the TFA under N₂ flow, the residue was dissolved in MeOH (1 mL), followed by slow addition of cold methyl tert-butyl ether (MTBE, 45 mL) to precipitate out a yellow product. The N-Boc deprotected polymer was further purified via three times of dissolution-precipitation process using the solvent of MeOH/MTBE (1

mL/45 mL) then vacuum dried and dissolved in Milli-Q water. Subsequently, the solution was subjected to lyophilization to give a final dye-labelled polymer (DM_{0.8i}Pen_{0.2})₂₀, which was used for confocal imaging.

Time-lapse fluorescent confocal imaging assay. The confocal imaging assay for drug-resistant *S. aureus* and drug-resistant *E. coli* was conducted according to the protocol in our previous study [79]. Briefly, the dye-labelled (DM_{0.8i}Pen_{0.2})₂₀ (4 × MBC, green fluorescence) and propidium iodide (40 mM, red fluorescence) were mixed in equal volumes to prepare a working solution. In addition, the bacteria were cultured in LB medium at 37 °C for 6 h to obtain the bacterial suspension, the bacterial suspension was washed by PBS buffer and then diluted in MH medium to achieve a working suspension with a cell density of 1 × 10⁷ CFU/mL. 10 μL of the bacterial suspension was dropped into a glass-bottomed cell culture dish for 10 minutes to allow the bacteria to attach to the bottom. Subsequently, 10 μL of working solution was added to the bacterial drop. The confocal images were captured at the various time points for three channels: bright field, 488 nm (green fluorescence) and 562 nm (red fluorescence), respectively. These images were used to record the bactericidal process.

Outer membrane permeabilization assay. The outer membrane permeabilization assay for drug-resistant *E. coli* was conducted according to the protocol in our previous study [42]. Briefly, the bacteria were cultured in LB medium at 37 °C for 6 h to obtain the bacterial suspension, the bacterial suspension was washed by PBS buffer and then diluted in HEPES medium (5 mM HEPES, 20 mM glucose, pH= 7.4) to achieve a working suspension with a cell density of 3 × 10⁸ CFU/mL, followed by addition of 1-N-phenyl-naphthylamine (NPN) dye at a final concentration of 10 μM. 90 μL of working suspension containing NPN was added to each well of a 384-well plate. The fluorescence changes (excitation λ= 350 nm, emission λ= 420 nm) were recorded on a SpectraMax® M2 plate reader (Molecular Devices, USA). Once the fluorescence intensity remained stable, 10 μL of (DM_{0.8i}Pen_{0.2})₂₀ was added to the bacterial solution, and the fluorescence intensity was recorded continuously.

Cytoplasmic membrane depolarization assay. The cytoplasmic membrane depolarization for drug-resistant *S. aureus* and drug-resistant *E. coli* was conducted according to the protocol in our previous study [42]. The drug-resistant bacteria were cultured in LB medium at 37 °C for 6 h, and then the bacterial suspension was diluted in HEPES medium (5 mM HEPES, 20mM glucose, pH= 7.4) to achieve a working suspension with a cell density of 1 × 10⁷ CFU/mL, followed by addition of 3, 3'-dipropylthiadicarbocyanine iodide (diSC3(5)) dye at a final concentration of 0.8 μM. The bacterial suspension was incubated for 1 h, followed by the addition of KCl to a final concentration of 0.1 M to balance the cytoplasmic and external K⁺ concentration. 90 μL of bacterial suspension containing diSC3(5) was added to each well of a 384-well plate. The fluorescence changes (excitation λ= 622 nm, emission λ= 673 nm) were recorded on a SpectraMax® M2 plate reader (Molecular Devices, USA). Once the fluorescence intensity remained stable, 10 μL of (DM_{0.8i}Pen_{0.2})₂₀ and 0.1% Triton X-100 as the positive control was separately added to the bacterial solution and the fluorescence intensity was recorded continuously.

Cytoplasmic membrane permeabilization assay. The cytoplasmic membrane permeabilization assay for drug-resistant *S. aureus* and drug-resistant *E. coli* was conducted according to the protocol in our previous study [42]. The drug-resistant bacteria were cultured in LB medium at 37 °C for 6 h, and then the bacterial suspension was diluted in HEPES medium (5 mM HEPES, 5 mM glucose, pH= 7.4) to achieve a working suspension with a cell density of 1 × 10⁸ CFU/mL, followed by addition of propidium iodide (PI) dye at a final concentration of 10 μM. 150 μL of bacterial suspension containing PI was added to each well of a corning 96-well plate. The fluorescence changes (excitation λ= 535 nm, emission λ= 617 nm) were recorded on a SpectraMax® M2 plate reader (Molecular

Devices, USA). Once the fluorescence intensity remained stable, 10 μL of $(\text{DM}_{0.8}\text{iPen}_{0.2})_{20}$ was added to the bacterial solution and the fluorescence intensity was recorded continuously.

SEM characterization of bacteria morphology. The SEM characterization for drug-resistant *S. aureus* and drug-resistant *E. coli* was conducted according to the protocol in our previous study [78]. Briefly, the drug-resistant bacteria were cultured in LB medium at 37 °C for 9 h, and then the bacterial suspension was diluted in LB medium to achieve a working suspension with a cell density of 1×10^7 CFU/mL, followed by addition of $(\text{DM}_{0.8}\text{iPen}_{0.2})_{20}$ at a final concentration of $1 \times \text{MBC}$. The bacterial suspension was incubated at 37 °C for 30 min. An untreated bacteria suspension was used as the control. $(\text{DM}_{0.8}\text{iPen}_{0.2})_{20}$ treated and untreated bacteria were collected by centrifugation at $1700 \times g$ for 5 min. They were washed with phosphate buffer saline (PBS) once and then fixed with 4% glutaraldehyde in phosphate buffer (PB) at 25 °C overnight. The bacteria were further washed with PBS and dehydrated with gradient ethanol (EtOH) solutions (30, 50, 70, 80, 90, 95, and then 100% ethanol). The samples were dried in air and then used for Field Emission Scanning Electron Microscopy (FESEM) characterization.”

Fig. 8 Antimicrobial mechanism study. **a)** Cytoplasmic membrane depolarization of $(DM_{0.8i}Pen_{0.2})_{20}$ against *S. aureus* USA300. **b)** Cytoplasmic membrane permeability of $(DM_{0.8i}Pen_{0.2})_{20}$ against *S. aureus* USA300. **c)** SEM characterization on *S. aureus* USA300 with and without $(DM_{0.8i}Pen_{0.2})_{20}$ treatment at $1 \times$ MBC concentration. **d)** Time-laps confocal fluorescence imaging on the interaction between *S. aureus* USA300 and fluorescent dye-labelled $(DM_{0.8i}Pen_{0.2})_{20}$ at $1 \times$ MBC concentration, in the presence of PI. **e)** Outer membrane permeability of $(DM_{0.8i}Pen_{0.2})_{20}$ against *E. coli*R19. **f)** Cytoplasmic membrane depolarization of $(DM_{0.8i}Pen_{0.2})_{20}$ against *E. coli*R19. **g)** SEM characterization on *E. coli*R19 with and without $(DM_{0.8i}Pen_{0.2})_{20}$ treatment at $1 \times$ MBC concentration. **h)** Time-laps

confocal fluorescence imaging on the interaction between *E. coli*R19 and fluorescent dye-labelled $(DM_{0.8}iPen_{0.2})_{20}$ at $1 \times$ MBC concentration, in the presence of PI.

Fig. S56 Cytoplasmic membrane permeabilization of $(DM_{0.8}iPen_{0.2})_{20}$ against *E. coli*R19.

Fig. S62 GPC trace of N-Boc protected dye-labelled polymer using DMF as the mobile phase.

4. *The synthesized polymer requires further structural characterization.*

Response: We greatly thank the reviewer for reminding of this point, which motivates us to improve the manuscript. By following the reviewers' suggestion, the MALDI-TOF-MS characterization of β -amino acid polymers $(DM_xiPen_y)_{20}$ were supplemented in the revised supporting information as shown below (new Supplementary Fig. S63-S68).

$$m/z: 161.1+128.1n+127.1+23.0$$

a: m/z = 2233 DM ₁₅	f: m/z = 2873 DM ₂₀
b: m/z = 2361 DM ₁₆	g: m/z = 3001 DM ₂₁
c: m/z = 2489 DM ₁₇	h: m/z = 3129 DM ₂₂
d: m/z = 2617 DM ₁₈	i: m/z = 3257 DM ₂₃
e: m/z = 2745 DM ₁₉	j: m/z = 3385 DM ₂₄

Fig. S63 MALDI-TOF MS characterization of antibacterial β -amino acid polymer DM₂₀.

$$m/z: 161.1+128.1x+141.1y+127.1+23.0$$

a: m/z = 2259 DM ₁₃ iPen ₂	g: m/z = 3027 DM ₁₉ iPen ₂
b: m/z = 2387 DM ₁₄ iPen ₂	h: m/z = 3155 DM ₂₀ iPen ₂
c: m/z = 2515 DM ₁₅ iPen ₂	i: m/z = 3283 DM ₂₁ iPen ₂
d: m/z = 2643 DM ₁₆ iPen ₂	j: m/z = 3411 DM ₂₂ iPen ₂
e: m/z = 2784 DM ₁₆ iPen ₃	k: m/z = 3539 DM ₂₃ iPen ₂
f: m/z = 2899 DM ₁₈ iPen ₂	l: m/z = 3664 DM ₂₄ iPen ₂

Fig. S64 MALDI-TOF MS characterization of antibacterial β -amino acid polymer (DM_{0.9}iPen_{0.1})₂₀.

$$m/z: 161.1+128.1x+141.1y+127.1+23.0$$

a: m/z = 1772 DM ₇ iPen ₄	h: m/z = 2682 DM ₁₃ iPen ₅
b: m/z = 1900 DM ₈ iPen ₄	i: m/z = 2810 DM ₁₄ iPen ₅
c: m/z = 2028 DM ₉ iPen ₄	j: m/z = 2938 DM ₁₅ iPen ₅
d: m/z = 2157 DM ₁₀ iPen ₄	k: m/z = 3066 DM ₁₆ iPen ₅
e: m/z = 2298 DM ₁₀ iPen ₅	l: m/z = 3207 DM ₁₆ iPen ₆
f: m/z = 2426 DM ₁₁ iPen ₅	m: m/z = 3322 DM ₁₈ iPen ₅
g: m/z = 2554 DM ₁₂ iPen ₅	n: m/z = 3477 DM ₁₇ iPen ₇

Fig. S65 MALDI-TOF MS characterization of antibacterial β -amino acid polymer (DM_{0.8}iPen_{0.2})₂₀.

Fig. S66 MALDI-TOF MS characterization of antibacterial β -amino acid polymer (DM_{0.7}iPen_{0.3})₂₀.

Fig. S67 MALDI-TOF MS characterization of antibacterial β -amino acid polymer (DM_{0.6}iPen_{0.4})₂₀.

Fig. S68 MALDI-TOF MS characterization of antibacterial β -amino acid polymer (DM_{0.5}iPen_{0.5})₂₀.

5. I would recommend the authors to explore and utilize additional machine learning models in the first stage, such as XGBOOST, to undertake this task for potentially enhanced results.

Response: We thank the reviewer for the suggestions. We have added additional three machine learning models, including XGB (*ACM SIGKDD*, **2016**, 785), GBDT (*Ann. Stat.*, **2011**, 29, 1189), Adaboost (*The Elements of Statistical Learning: Data Mining, Inference, and Prediction*, **2009**, New York: Springer), to comprehensively evaluate the performance of applying descriptor downselection and data augmentation. Generally speaking, for all different machine models, applying descriptor downselection and data augmentation brings positive effects and GBDT performs best among all the models, better than RF which was originally used. We conclude all the results in new Fig. 2, and we make corresponding revision in the manuscript.

“In this manuscript, we randomly selected 80% of collected 86 polymers as the training set D_{train_ori} and the rest of 20% of data were set as the unseen testing set D_{test} . Thus, all models were trained and evaluated in same data situation. We first evaluated the performance of applying descriptor downselection and data augmentation that were two important operations of influencing the input representations. We defined an augmented training data D_{train_aug} , which contained original training data D_{train_ori} along with additional data by tuning all possible polymer sequences of cationic and hydrophobic subunits in all representations. In this stage, we constructed 4 classic machine learning based regression models (including GBDT [47], RF [45], XGB [48] and Adaboost [49]) for bioactivity prediction. The model performance was characterized by calculating the mean R-squared coefficient (R2). We applied a 15-fold cross validation on D_{train_ori} and D_{train_aug} to evaluate the performance of different models with fixed descriptors (Fig. 2a-2l and Fig. S9).

Generally speaking, GBDT models performed best than other methods on each task (Fig. 2a-2c for GBDT, Fig. 2d-2f for RF, Fig. 2g-2i for XGB, Fig. 2j-2l for Adaboost). The results showed that the mean R2 values of GBDT for D_{train_ori} increased gradually to the 0.626, 0.640 and 0.795 on predicting the values of $MIC_{S. aureus}$, $MIC_{E. coli}$ and HC_{10} of polymers with applying descriptors downselection, showing that more related information was selected step by step (Fig. 2a-2c, blue boxes). After applying data augmentation, the mean R2 values showed a more obvious increase to 0.739, 0.681 and 0.831 for D_{train_aug} compared to using D_{train_ori} , indicating the increased prediction accuracy. It was worth noting that the error bar values of R2 also significantly decreased for D_{train_aug} which signified an improvement in the stability for prediction (Fig. 2a-2c, red boxes). Via a final evaluation with GBDT on D_{test} , the mean R2 values reached 0.672, 0.537 and 0.834 for $MIC_{S. aureus}$, $MIC_{E. coli}$ and HC_{10} , regarding as a machine learning baseline in this manuscript. Results for all machine learning models on D_{test} were demonstrated in Fig. 2m-2o.”

Fig 2 a-l) Cross validation results with GBDT (a-c), RF (d-f), XGB (g-i), Adaboost (j-l) for applying descriptors downselection and data augmentation on the value of MIC_{S. aureus}, MIC_{E. coli} and HC₁₀. “Descriptor_Init to Descriptor_Opt” are different sets of descriptors when downselection. The borders of the boxes indicate the first quartile (left) and the third quartile (right) of the results. The line in the box indicates the median. The whiskers refer to the most extreme, nonoutlier data points.

6. In Fig. 2, the best result in a does not align with those in d, and b exhibits the same issue.

Response: We thank the reviewer for the question. As described in the manuscript, we randomly selected 80% of collected 86 polymers as the training set D_{train_ori} and the rest of 20% of data were set as the unseen testing set D_{test} . The results in new Fig. 2a-2i (corresponded to Fig. 2a-2c in previously submitted version) were all the cross-validation results using the original data D_{train_ori} and the augmented data D_{train_aug} , aiming to prove the effectiveness applying descriptor downselection and data augmentation.

After training all machine learning based predictive model and neural network based (ours) predictive model, all these models were evaluated on the unseen data D_{test} , which were not used in the cross-validation stage. All the results were shown in new Fig. 2m-2o (corresponded to Fig. 2d-2f in previously submitted version). In this way, these two parts of results would not align together since the evaluated data were not same.

In this revision, we add all machine learning based predictive results as baseline in new Fig. 2. Simultaneously, we retrain all the models with more training parameters and random seeds for more comprehensive results. We adjust partial descriptions to avoid misleading and we repaint all the figures. The revision is shown below.

“Moreover, we further studied the performance of all the predictive network by combing three modals of text sequence of polymer, polymer graph and descriptors with applying descriptor downselection and data augmentation discussed before. In addition, we added GBDT, RF, XGB and

Adaboost as basic benchmark models and we also introduce the most commonly used polymer representation of Morgan fingerprints [50, 51] for comparison. All models were trained on D_{train_aug} and evaluated on D_{test} for performance comparison, and R2 was again used as the metric. We designed different deep neural network structures for each single representation and an integrated framework for multi-modal representations (see Methods). With final evaluation on D_{test} , it is obviously found that GBDT again demonstrated the best in all machine learning based models with mean R2 values of 0.672, 0.537 and 0.834 for $MIC_{S. aureus}$, $MIC_{E. coli}$ and HC_{10} (Fig. 2m-2o). The “Descriptor_Opt” demonstrated the best in all single representation with mean R2 values of 0.606, 0.415 and 0.852, whereas the mean R2 values of Morgan was 0.606, 0.415 and 0.852. In addition, the combination of three modals “Seq+Graph+Descriptor_Opt” showed the highest mean R2 values at 0.697, 0.556 and 0.900, indicating that our constructed multi-modal polymer representations obviously improved the accuracy and stability of the predictive model for few-shot polymers (More results concluded in Fig. S13 and Table S3-S4).”

Fig. 2 m-o) Property prediction results of the unseen test set D_{test} with baseline machine learning models and deep neural network on $MIC_{S. aureus}$, $MIC_{E. coli}$ and HC_{10} with different polymer representation combination. “Seq” is the abbreviation of “Sequence”.

7. Equation (5) should be elaborated upon in greater detail, and within the “predictive network” section, it is essential to explicitly specify the dimensional changes of the variables within the model.

Response: We thank the reviewer for the suggestion and we have added more details about the dimensional changes of the variables as shown below.

“...The dimensionality of the input feature F_j is $[B, N_D]$ and the dimensionality of the input layer of FNN is $[N_D, D_f]$, where B is the number of batch size, N_D is the number of descriptors used and D_f is the dimensionality of the hidden layers in FNN. For 2)...”

“...The dimensionality of the input sequence vector is $[B, n]$ and the dimensionality of the sequence embedding is $[n, D_s]$, where B is the number of batch size, n is the number of each input sequence and D_s is the dimensionality of the hidden layers in GRU. The final dimensionality of the sequence feature F_s is $[B, D_s]$...”

“For 3), we apply a Bidirectional Message Communication GNN [59], which make full use of the node message for more effective message interactions to extract the local information embedded in the graph. The network structures can be seen in Fig. S11 and the pseudocode of the model were concluded in supplementary information as Algorithm 1.

Specifically, the input of the algorithm is each polymer graph $G = (\mathcal{V}, \mathcal{E})$ and all of its atom attributes $x_v (\forall v \in \mathcal{V})$ and bond attributes $x_{e_{vw}} (\forall e_{vw} \in \mathcal{E})$. The initial node feature h_v^0 is simply the atom attributes, while the initial edge feature $h_{e_{vw}}^0$ is the bond attributes. Then, according to the network depth T , a T steps message aggregation and update procedure is applied. In each step t , each node message vector m_v^{t+1} is aggregated according to its incoming edges and each edge message vector $m_{e_{vw}}^{t+1}$ is aggregated according to its neighbour nodes, shown as,

$$\begin{cases} m_v^{t+1} = \text{MAX}(h_{e_{uv}}^t) \odot \text{SUM}(h_{e_{uv}}^t), u \in \mathcal{N}(v), \\ m_{e_{vw}}^{t+1} = \text{MEAN}(h_v^t, h_w^t), \end{cases} \quad (5)$$

where **MAX**, **SUM**, **MEAN** are the corresponding aggregating strategy, \odot is an element-wise multiplication operator. Then the obtained message vectors of node and edge $m_v^{t+1}, m_{e_{vw}}^{t+1}$ are concatenated with the corresponding current hidden states to be sent to the communicate function which use an addition operator as communicative kernel to calculate the communicative vector $p_v^{t+1}, p_{e_{vw}}^{t+1}$. Then the hidden state of the node and edge are updated with skip connection as,

$$\begin{cases} h_v^{t+1} = \text{ReLU}(h_v^0 + \mathbf{W}_v \cdot p_v^{t+1}), \\ h_{e_{vw}}^{t+1} = \text{ReLU}(h_{e_{vw}}^0 + \mathbf{W}_e \cdot p_{e_{vw}}^{t+1}), \end{cases} \quad (6)$$

where **ReLU** is the rectified linear unit and $\mathbf{W}_v, \mathbf{W}_e$ are learned matrices. After T step iteration, a GRU based readout function is applied to the final node representation h_v^T to get the graph-level representation F_g as,

$$F_g = \sum_{v \in \mathcal{V}} \text{GRU}(h_v^T). \quad (7)$$

The dimensionality of the input atom vector and bond vector in graph are $[B, N_v, F_v]$ and $[B, N_e, F_e]$, and the dimensionality of the atom embedding and bond embedding in Bidirectional Message Communication GNN are $[F_v, D_g]$ and $[F_e, D_g]$, where B is the number of batch size, N_v, N_e are the atom number and bond number in each input molecular graph, F_v and F_e are the number of attributes for each atom and bond and D_g is the dimensionality of the hidden layers in GNN. The final dimensionality of the graph feature F_g is $[B, D_g]$. For 1)-3)...”

“...where **Combine** was the function to assemble different representations with adding and stacking, and F_j is the joint feature by stacking all the vector features with the dimensionality of $[B, D_j]$, $D_j = D + N_D (D = D_f = D_s = D_g)$. According to..”

“... of keys. With the calculation of the Transformer block, we got the final predictions of the properties P with the dimensionality of $[B, 1]$,

$$P = \text{FNN}(\text{Transformer}(F_j, F_f)). \quad (11)$$

8. *The authors should elucidate the novelty in structure of the polymers selected for experimentation in comparison to previously discovered materials.*

Response: We thank the reviewer for the valuable suggestions. Following the reviewer’s suggestion, we have supplemented the description about the novelty in structure of the polymers selected for experimentation in the revised manuscript as shown below.

“...It was worth noting that DM_{0.8}/Pen_{0.2} possessed a unique structure characterized by different hydrophobic subunits in comparison to previously reported antimicrobial β -amino acid polymers.

Importantly, our prediction showed that $DM_{0.8i}Pen_{0.2}$ had lower cytotoxicity and improved antimicrobial selectivity compared to amphiphilic polymers reported earlier, which utilized DM as the cationic subunit and hydrophobic subunits with lower carbon numbers, such as DM:CHx, DM: β CP, DM: β CH [41].”

9. *Given the limited amount of data, the authors should provide a detailed description of the training process for the predictive model, as well as the methodology employed to select the optimal model to prevent overfitting.*

Response: We thank the reviewer for the suggestion and we have added more description about the training process and training settings for both the predictive model and generative model and how to choose the model as shown below.

“*Predictive Model Training Settings.* We randomly split the training data D_{train_aug} into 8:1:1 train/valid/test ratios and we applied bayesian optimization to find the optimal hyper-parameters. Then we used the optimized parameters to retrain the model for 10 independent runs with different random seeds. Specifically, a dynamic changed learning rate was used with the Adam optimizer with Mean Squared Error (MSE) loss to train the model. We set an initial learning rate as 10^{-4} and it would be doubled as a max learning rate in 5 warm up training epochs, and finally the learning would return the 10^{-4} as a final value. The training epoch and the batch size were set as 100 and 16 respectively. In each epoch, if the validation MSE reduced, the model would be saved. The parameters of each block of GNN, GRU, Transformer and FNN were all recorded in Table S2. In addition, since the operation of random data splitting would cause uneven distribution of training data, we applied the ensembling technique, which is a common technique in machine learning. Multiple independently trained models with different random seeds were combined to produce an averaging prediction so as to prevent overfitting on partial results. After training, the unseen testing data D_{test} is used to evaluate the performance of the model, using the R-square (R2, higher R2 means better performance of the model) and Root-Mean-Squared Error (RMSE, lower RMSE means better performance of the model) as metrics. The implementation of the model relies on Pytorch and RDKit package.”

“*Generative Model Training settings.* For the graph grammar distillation pre-training process, the training epoch, batch size and the learning rate were set as 500, 256 and 10^{-3} respectively. The dimensionality of hidden layers of GRU was set as 256. For the reinforcement learning fine-tuning process, the training epoch, batch size and the learning rate were set as 500, 30 and 10^{-9} respectively. Also, we use the Negative Log Likelihood (NLL) loss to train the model and the implementation of the model relies on Pytorch and RDKit package. All hyperparameters are concluded in Table S5.”

Reviewer 2:

Comments:

The accurate prediction of antimicrobial polymers composed of unnatural amino acids is a long-term challenge due to limited polymer datasets. The manuscript describes the striking advance in accurate prediction and discovering on HDP-mimicking β -amino acid polymers via a few-shot polymer inverse design framework. The authors successfully identify 83 optimal antibacterial polymers via simulating prediction of over 10^5 polymers. This is a very interesting and important paper in the field of research on the discovery of antibacterial compounds against drug-resistant bacteria. In addition, the author synthesized an optimal polymer and found that the polymer exhibited broad-spectrum and potent antibacterial activity against multiple clinically isolated antibiotic-resistant pathogens, validating the effectiveness of AI-guided design strategy. In summary, the reviewer suggests the acceptance of the article for publication in Nature Communications after addressing the minor points.

Response: We thank the reviewer for the favorable comments and the detailed questions below.

1. *It is recommended to add the description of the importance of research on antimicrobial β -amino acid polymers and provide references to the latest literature.*

Response: We thank the reviewer for reminding us on this and we have supplemented description of the importance of research on antimicrobial β -amino acid polymers and added related references in our revised manuscript as shown below.

“...HDP-mimicking β -amino acid polymers have attracted significant attention and demonstrated enormous potential for various applications due to striking structural similarity to natural peptides, superior biocompatibility and high resistance to protease hydrolysis [13, 35-37]...”

2. *Mass spectrometric characterization of positively charged monomer DM is suggested to be supplemented.*

Response: We thank the reviewer for reminding us on this. The normal mass spectrometric characterization of positively charged monomer DM were added into in our revised manuscript as shown below.

“ β -lactam DM was synthesized by following previously reported procedure to obtain the pure product as a white solid in 42.6% overall yield. $^1\text{H NMR}$ (600 MHz, CDCl_3 , see Fig. S54): δ 5.9 (s, 1H), 4.9 (s, 1H), 3.66-3.56 (m, 1H), 3.28 (t, $J = 10.2$ Hz, 1H), 2.97 (t, $J = 7.8$ Hz, 1H), 1.45 (s, 3H), 1.43 (s, 9H), 1.37 (s, 3H). ESI-MS: m/z calculated for $\text{C}_{11}\text{H}_{21}\text{N}_2\text{O}_3$ $[\text{M}+\text{H}]^+$: 229.2; Found 229.2.”

3. *Both x , y and $x\%$, $y\%$ are used to present the ratio of the two subunits of β -amino acid polymers. The ratio of positively charged subunits to hydrophobic subunits should be standardized.*

Response: We thank the reviewer for the pointing out the problem and we have standardized all ratio information $x\%$, $y\%$ into corresponding x , y , shown in new Fig. 1, Fig. 6 and all supplementary materials (new Fig. S1, Fig. S10 and Table S6-S22).

Fig. 1 Framework overview. **a)** By collecting 86 data comprising chemical structures and their bioactivity of β -amino acid polymers, we develop an AI-guided polymer design framework to find promising polymers with broad-spectrum antibacterial efficacy and low cytotoxicity. In addition, we conduct a refined classification according to the different position of side chain substituents and cyclic or non-cyclic substitution pattern, which defines a scaffold set for the following polymer generation. x and y are defined as the percentages of a positively charged subunit and a hydrophobic subunit in β -amino acid polymers, respectively. **b)** We conduct a multi-modal polymer representation method, including text sequence, graph with additional polymer settings and descriptors embedded with 2D- and 3D-properties of subunits to expand for multi-scale polymer information to realize few-shot polymer prediction.

Fig. 6 Discovery of broad-spectrum antibacterial polymers bearing β^3 -amino acid. **a**) Various β^3 -amino acids generated in the discovery process with fixed DM subunit. Note that all subunits are achiral. x and y are defined as the percentages of a positively charged subunit and a hydrophobic subunit in β^3 -amino acid polymers, respectively. **b**) 3D-projection of the bioactivities on MIC_{S. aureus}, MIC_{E. coli} and HC₁₀ for the generated β^3 -amino acid polymers with fixed DM subunit. Orange plots are the projection on (HC₁₀, MIC_{S. aureus}) space, while green plots are the projection on (HC₁₀, MIC_{E. coli}) space. Red stars are polymers reaching three desired properties of MIC_{S. aureus} < 25, MIC_{E. coli} < 25 and HC₁₀ > 100, simultaneously. **c**) Display of the final filtered out candidate polymers and we choose Candidate 5 for further synthesis and bioactivity validation.

4. There is an error in the table of Fig 5, the table formatting should be corrected and the abbreviations should be given full descriptions.

Response: We thank the reviewer for reminding us on this. We have corrected the table formatting and adjusted the previous abbreviations “Cati.:Hydr.” into “x:y” in new Fig 6 (corresponding to the previous Fig 5) as shown below.

c

Structure	Polymer	x:y	HC ₁₀ (μg/mL)	MIC _{S. aureus} (μg/mL)	MIC _{E. Coli} (μg/mL)
	Candidate 1	9:1	119.6	8.31	24.9
	Candidate 2	7:3	134.6	15.5	14.6
	Candidate 3	8:2	137.7	13.9	15.2
	Candidate 4	9:1	214.8	12.7	20.0
	Candidate 5	8:2	156.9	10.2	24.1

Fig. 6 c) Display of the final filtered out candidate polymers and we choose Candidate 5 for further synthesis and bioactivity validation.

5. The definition of the end group R in the chemical structure of antibacterial β -amino acid polymers in Fig 6 should be supplemented.

Response: We thank the reviewer for pointing out this and have supplemented the definition of the end group R in the revised manuscript as shown below.

Fig. 7 Experimental validation. a) Synthesis of host defense peptides-mimicking β -amino acid polymers DM_xiPen_y ($x+y=1$, $y=0-0.5$), R represents the side chain from one of the starting monomers of DM and iPen.

6. The author should check the writing standardization and labeling of chemical structures of polymers throughout the manuscript, such as heterochiral structures (\pm), superscripts in ¹³C NMR, etc.

Response: We thank the reviewer for pointing out this and we have checked and standardized the chemical structures throughout the manuscript in the revised manuscript.

REVIEWERS' COMMENTS

Reviewer #2 (Remarks to the Author):

I have carefully reviewed the revised version of the manuscript. The modifications and additions to the manuscript have significantly improved the clarity, depth, and robustness of the study.

Reviewer #2 (Remarks on code availability):

I have thoroughly examined the code associated with the manuscript. The repository includes a comprehensive README file that provides detailed instructions for installation, configuration, and operation of the codebase.

Reviewer #4 (Remarks to the Author):

Comments are addressed well. I have not more concerns. The manuscript is acceptable at its current format.